# DenseGRPO: From Sparse to Dense Reward for Flow Matching Model Alignment

**Haoyou Deng**[1,2*], **Keyu Yan**[2*], **Chaojie Mao**[2], **Xiang Wang**[1], **Yu Liu**[2], **Changxin Gao**[1], **Nong Sang**[1†]

[1]Key Laboratory of Image Processing and Intelligent Control,
School of Artificial Intelligence and Automation, Huazhong University of Science and Technology
[2]Tongyi Lab, Alibaba Group

{haoyoudeng, wxiang, cgao, nsang}@hust.edu.cn
yankeyu66@aliyun.com {chaojie.mcj, ly103369}@alibaba-inc.com

## Abstract

Recent GRPO-based approaches built on flow matching models have shown remarkable improvements in human preference alignment for text-to-image generation. Nevertheless, they still suffer from the sparse reward problem: the terminal reward of the entire denoising trajectory is applied to all intermediate steps, resulting in a mismatch between the global feedback signals and the exact fine-grained contributions at intermediate denoising steps. To address this issue, we introduce **DenseGRPO**, a novel framework that aligns human preference with dense rewards, which evaluates the fine-grained contribution of each denoising step. Specifically, our approach includes two key components: (1) we propose to predict the step-wise reward gain as dense reward of each denoising step, which applies a reward model on the intermediate clean images via an ODE-based approach. This manner ensures an alignment between feedback signals and the contributions of individual steps, facilitating effective training; and (2) based on the estimated dense rewards, a mismatch drawback between the uniform exploration setting and the time-varying noise intensity in existing GRPO-based methods is revealed, leading to an inappropriate exploration space. Thus, we propose a reward-aware scheme to calibrate the exploration space by adaptively adjusting a timestep-specific stochasticity injection in the SDE sampler, ensuring a suitable exploration space at all timesteps. Extensive experiments on multiple standard benchmarks demonstrate the effectiveness of the proposed DenseGRPO and highlight the critical role of the valid dense rewards in flow matching model alignment.

## 1 Introduction

Flow matching models (Lipman et al., 2022; Liu et al., 2022; Wang et al., 2025a) have achieved remarkable advancement in the text-to-image generation task, yet aligning them with human preference remains a critical challenge. Recent progresses (Liu et al., 2025; Xue et al., 2025; Wang et al., 2025b) highlight reinforcement learning (RL) as a promising solution by maximizing rewards during the post-training stage. Among these, Group Relative Policy Optimization (GRPO) (Shao et al., 2024) has attracted substantial attention, with numerous studies (Liu et al., 2025; Xue et al., 2025; Li et al., 2025; He et al., 2025) reporting significant gains in human preference alignment.

Although effective, existing GRPO-based approaches, *e.g.*, Flow-GRPO (Liu et al., 2025) and DanceGRPO (Xue et al., 2025), still suffer from the sparse reward problem: the terminal reward of the entire denoising trajectory is directly adopted to optimize intermediate denoising steps. As shown in Fig. 1 (a), for the $i$-th $T$-step generation trajectory in a GRPO sampled group, they only predict a single, sparse reward $R^i$ from the terminal generated image, and naively adopt $R^i$ to optimize all intermediate denoising steps. However, as $R^i$ represents the cumulative contribution of all

---

[*]Equal Contribution   [†]Corresponding Author

$T$ denoising steps, directly applying $R^i$ to optimize a single step at $\text{timestep} = t$ leads to a mismatch between the assigned global trajectory-level feedback and the exact fine-grained step-wise contribution, misleading policy optimization.

To address the aforementioned issue, we introduce DenseGRPO, a novel RL framework that aligns human preference with dense rewards, as depicted in Fig. 1 (b). The key idea of dense rewards is to evaluate the step-wise contribution of each denoising step, thereby aligning the feedback signals with the fine-grained contribution. Intuitively, training a process reward model presents a promising approach to estimate dense rewards (Zhang et al., 2024), yet it encounters two limitations: increased training costs due to additional models and limited adaptability to other tasks. In DenseGRPO, we adopt a simple yet effective approach that eliminates the need for additional specialized models and can seamlessly integrate with any established reward model. Specifically, since the contribution of a denoising step can be accessed by latent change, we propose to predict the reward gain between the current step and the next step latent as dense reward of each denoising step. To estimate the reward of an intermediate latent, we leverage the deterministic nature of Ordinary Differential Equation (ODE) and apply a reward model on the intermediate clean images via ODE denoising. Then, we assign the reward feedback as latent rewards and thus obtain the dense rewards by computing reward gains at each step. By this means, the estimated dense rewards ensure an alignment between feedback signals and the contribution of individual steps, thereby facilitating human preference alignment.

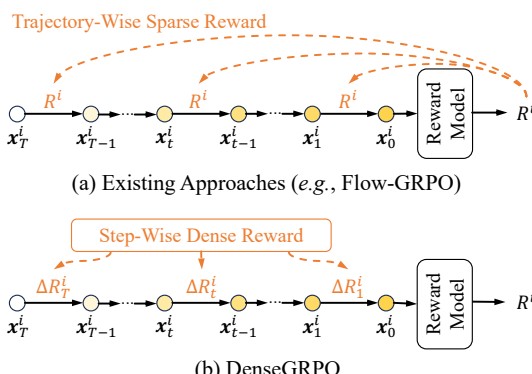

(a) Existing Approaches (*e.g.*, Flow-GRPO)

(b) DenseGRPO

Figure 1: (a) Existing approaches only predict a single, sparse reward at the end of the denoising trajectory, which is naively applied to optimize all intermediate steps. (b) DenseGRPO estimates step-wise rewards of individual steps, densifying the feedback signal for the denoising process.

Moreover, leveraging the step-wise dense rewards estimated above, a mismatch drawback between the uniform exploration setting and the time-varying noise intensity in existing GRPO-based approaches (Liu et al., 2025; Xue et al., 2025) is revealed. In general, the amount of noise is consistent between the diffusion and denoising processes in diffusion models (Song et al., 2020). Since RL relies on stochastic exploration, Flow-GRPO proposes a Stochastic Differential Equation (SDE) sampler that relaxes this consistency and injects increased noise, allowing diverse sampling. However, the current uniform setting of noise injection fails to align with the time-varying nature of the generation process, often resulting in either excessive or insufficient stochasticity. As evidenced by Fig. 3 (a), where nearly all samples receive negative rewards at late timesteps, the distribution of dense rewards is imbalanced, indicating an inappropriate exploration space. To mitigate this, we propose a reward-aware scheme to calibrate the exploration space by adaptively adjusting a timestep-specific stochasticity injection in the SDE sampler, ensuring a suitable exploration space for effective GRPO learning. We conduct extensive experiments across multiple benchmarks, and the superior performance of DenseGRPO demonstrates its effectiveness and underscores the critical role of valid dense rewards in flow matching model alignment.

To summarize, the main contributions of our work are as follows:

- We introduce DenseGRPO, which aligns human preference with dense reward, evaluating the fine-grained contribution of each denoising step. Leveraging an ODE-based approach, DenseGRPO estimates a reliable step-wise dense reward that aligns with the contribution.

- Informed by the estimated dense rewards, we propose a reward-aware scheme to calibrate the exploration space, balancing the dense reward distribution at all timesteps.

- Comprehensive experiments on multiple text-to-image benchmarks demonstrate the state-of-the-art performance of the proposed DenseGRPO and highlight the critical role of dense rewards in flow matching model alignment.

## 2 RELATED WORK

**Alignment for Text-to-Image Generation.** Aligning text-to-image models (Wei et al., 2025) with human preferences has attracted considerable attention. Early works are directly driven by the preference signals with scalar rewards (Prabhudesai et al., 2023; Xu et al., 2023) or reward weighted regression (Lee et al., 2023; Furuta et al., 2024). To obviate the need for a reward model, some approaches (Wallace et al., 2024; Yang et al., 2024a) adopt offline Direct Preference Optimization (DPO) (Rafailov et al., 2023) with win-lose pairwise data to directly learn from human feedback. In parallel, to tackle the distribution shift induced by offline win–lose pairwise data relative to the policy model during training, several methods (Black et al., 2023; Fan et al., 2023) utilize Proximal Policy Optimization (PPO) (Schulman et al., 2017) for online reinforcement learning, optimizing the score function through policy gradient methods. More recently, Group Relative Policy Optimization (GRPO) (Shao et al., 2024) has further improved the alignment task. Specifically, pioneering efforts, *e.g.*, Flow-GRPO (Liu et al., 2025) and DanceGRPO (Xue et al., 2025), introduce the GRPO framework on flow matching models and enable diversity exploration by converting the deterministic ODE sampler into an equivalent SDE sampler. Despite subsequent GRPO-based advances (He et al., 2025; Li et al., 2025; Wang et al., 2025b), existing methods still exhibit a mismatch between the global terminal reward feedback and exact fine-grained contribution at each denoising step, thereby limiting performance. To tackle this issue, we propose DenseGRPO that estimates and assigns accurate reward signals for each denoising step, thereby facilitating effective optimization.

**Dense Reward.** In sequential generation model alignment, dense reward has proven effective in addressing the sparse reward issue, which is inherent in the trajectory-level feedback. In text generation, to densify the sparse reward, several methods incorporate a per-step KL penalty into the training objective (Ramamurthy et al., 2022; Castricato et al., 2022). Additionally, Tan & Pan (2025) dynamically weights the rewards using token-level entropy for dense reward prediction, achieving true token-level credit assignment within GRPO framework. Similarly, dense reward has been explored for training text-to-image generation models. Specifically, within DPO-style methods, Yang et al. (2024b) fines the per-step reward signal and introduces temporal discounting into the training objective, and SPO (Liang, 2024) trains a step-aware performance model for both noise and clean images. In PPO-style approaches, Zhang et al. (2024) assigns each intermediate denoising timestep a temporal reward by learning a temporal critic function. Besides, TempFlow-GRPO (He et al., 2025) proposes a trajectory branching mechanism that provides per-timestep reward in GRPO-based alignment, yet adopts a trajectory-wise signal for step optimization. Most closely related to our work, CoCA (Liao et al., 2025) estimates the contribution of each step by assigning the terminal reward in proportion to the latent similarity. However, it still assigns the trajectory-wise reward signals to optimize an intermediate denoising step, where optimization mismatch persists. In contrast, we present DenseGRPO that aims to train with the step-wise dense reward, which captures the exact fine-grained contribution of each denoising step.

## 3 PRELIMINARY

In this section, we briefly review some concepts from a typical previous work (Liu et al., 2025) to provide preliminary details about the application of GRPO in flow matching models, including (1) the formulation of RL on flow matching models, (2) the GRPO framework, and (3) the SDE sampler.

**RL on Flow Matching Models.** Within reinforcement learning, a sequential decision-making problem is commonly formulated as a Markov Decision Process (MDP). An MDP is characterized by a tuple $(\mathcal{S}, \mathcal{A}, \rho_0, P, \mathcal{R})$, where $\mathcal{S}$ denotes the state space, $\mathcal{A}$ represents the action space, $\rho_0$ is the distribution of initial states, $P$ is the transition kernel, and $\mathcal{R}$ is a reward function. At timestep $t$ with a state $\mathbf{s}_t \in \mathcal{S}$, the agent takes an action $\mathbf{a}_t \in \mathcal{A}$ according to a policy $\pi(\mathbf{a} \mid \mathbf{s})$, and thereby receives a reward $R(\mathbf{s}_t, \mathbf{a}_t)$, moving to a new state $\mathbf{s}_{t+1} \sim P(\mathbf{s}_{t+1} \mid \mathbf{s}_t, \mathbf{a}_t)$. Following Flow-GRPO (Liu et al., 2025), the iterative denoising process in flow matching models can be formulated as an MDP:

$$\mathbf{s}_t \triangleq (\boldsymbol{c}, t, \boldsymbol{x}_t), \quad \pi(\mathbf{a}_t \mid \mathbf{s}_t) \triangleq p(\boldsymbol{x}_{t-1} \mid \boldsymbol{x}_t, \boldsymbol{c}), \quad P(\mathbf{s}_{t+1} \mid \mathbf{s}_t, \mathbf{a}_t) \triangleq (\delta_{\boldsymbol{c}}, \delta_{t-1}, \delta_{\boldsymbol{x}_{t-1}})$$

$$\mathbf{a}_t \triangleq \boldsymbol{x}_{t-1}, \quad R(\mathbf{s}_t, \mathbf{a}_t) \triangleq \begin{cases} \mathcal{R}(\boldsymbol{x}_0, \boldsymbol{c}), & \text{if } t = 0 \\ 0, & \text{otherwise} \end{cases}, \quad \rho_0(\mathbf{s}_0) \triangleq (p(\boldsymbol{c}), \delta_T, \mathcal{N}(\mathbf{0}, \mathbf{I})) \quad (1)$$

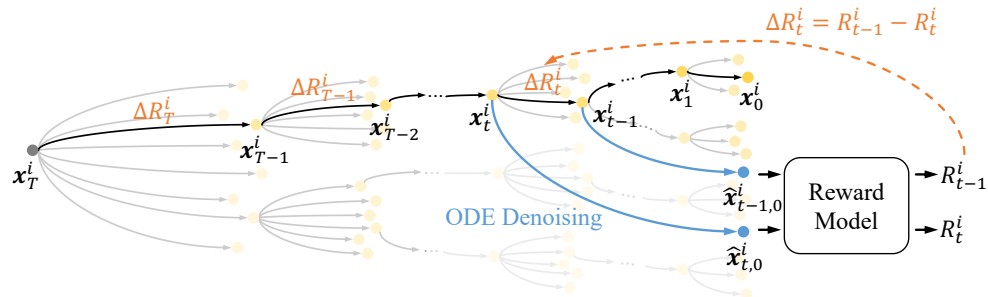

Figure 2: Overview of DenseGRPO. Given the $i$-th trajectory within a GRPO group, we first predict the rewards $\{R_t^i\}$ of latents $\{x_t^i\}$ via ODE denoising. By capturing the reward gain $\{\Delta R_t^i\}$ at each step, we obtain the dense reward that reliably evaluates the step-wise contribution.

Here, the state at timestep $t$ includes the prompt $c$, the timestep $t$, and the latent $x_t$. The action is the $x_{t-1}$ predicted by policy model $p(x_{t-1} \mid x_t, c)$. And $\delta_y$ is the Dirac delta distribution with nonzero density only at $y$. Notably, the reward acts as a trajectory-wise feedback signal that predicts a single, sparse reward only at the terminal state and provides zero reward at intermediate steps. This formulation assigns the reward of the entire trajectory to the final denoising step, thereby overlooking the fine-grained contributions of intermediate steps. As a result, existing methods adopt this sparse reward to optimize all timesteps, leading to a feedback-contribution mismatch. To address this, DenseGRPO explicitly estimates the step-wise dense rewards, thereby aligning the reward feedback with the contribution at each step.

**GRPO Framework.** Flow-GRPO adopts the GRPO framework (Shao et al., 2024) to align flow matching models. Specifically, given a prompt $c$, the flow matching model $p_\theta$ samples a group of $G$ individual images $\{x_0^i\}_{i=1}^G$ with $T$ timesteps and the corresponding denoising trajectories $\{(x_T^i, x_{T-1}^i, ..., x_0^i)\}_{i=1}^G$. Using a reward model $\mathcal{R}$, the advantage of the $i$-th image is estimated by group normalization as follows:

$$\hat{A}_t^i = \frac{\mathcal{R}(x_0^i, c) - \text{mean}(\{\mathcal{R}(x_0^i, c)\}_{i=1}^G)}{\text{std}(\{\mathcal{R}(x_0^i, c)\}_{i=1}^G)}. \tag{2}$$

Subsequently, the policy is optimized by maximizing the following objective:

$$\mathcal{J}_{\text{Flow-GRPO}}(\theta) = \mathbb{E}_{c \sim \mathcal{C}, \{x^i\}_{i=1}^G \sim \pi_{\theta_{\text{old}}}(\cdot|c)} f(r, \hat{A}, \theta, \epsilon, \beta), \tag{3}$$

where

$$f(r, \hat{A}, \theta, \epsilon, \beta) = \frac{1}{G}\sum_{i=1}^G \frac{1}{T}\sum_{t=0}^{T-1} (min(r_t^i(\theta)\hat{A}_t^i, \text{clip}(r_t^i(\theta), 1-\epsilon, 1+\epsilon)\hat{A}_t^i) - \beta D_{KL}(\pi_\theta||\pi_{\text{ref}})), \tag{4}$$

with $r_t^i(\theta) = \frac{p_\theta(x_{t-1}^i|x_t^i, c)}{p_{\theta_{\text{old}}}(x_{t-1}^i|x_t^i, c)}$. Notably, the advantage $\hat{A}_t^i$ obtained via Eq. 2 is exclusively determined by the reward signal $\mathcal{R}(x_0^i, c)$ of the entire trajectory, rendering it independent of any particular timestep $t$. In other words, policy optimization across different timesteps utilizes identical trajectory-wise reward feedback, exhibiting a mismatch between the assigned trajectory-wise feedback and the step-wise contributions of each timestep.

**SDE Sampler.** Typically, flow matching models predict the velocity $v_t$ and employ a deterministic ODE for the denoising process:

$$dx_t = v_t dt. \tag{5}$$

Yet, GRPO requires stochastic sampling to generate diverse trajectories for exploration. To this end, Flow-GRPO injects additional noise to sampling by converting the deterministic ODE sampler to an equivalent SDE sampler:

$$x_{t+\Delta t} = x_t + [v_\theta(x_t, t) + \frac{\sigma_t^2}{2t}(x_t + (1-t)v_\theta(x_t, t))]\Delta t + \sigma_t\sqrt{\Delta t}\epsilon. \tag{6}$$

Here, $\sigma_t = a\sqrt{\frac{t}{1-t}}$ and $\epsilon \sim \mathcal{N}(0, I)$ inject stochasticity, where $a$ is a scalar hyper-parameter for noise level control.

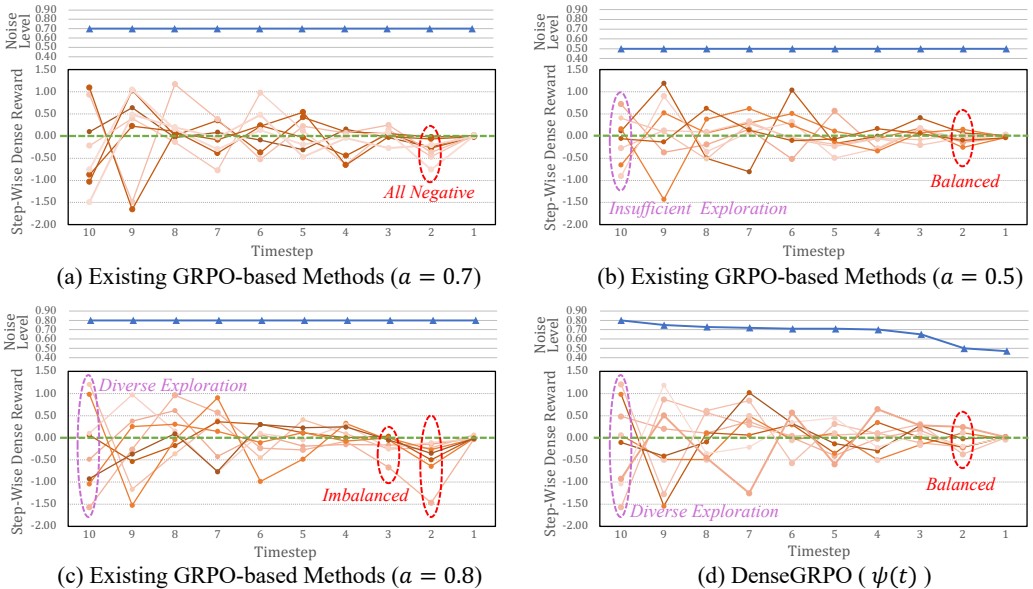

Figure 3: Visualization of dense rewards, where each polyline denotes an SDE-sampled trajectory: (a)(b)(c) existing GRPO-based methods utilize a uniform setting of noise level $a$, such as $a = 0.7$, $a = 0.5$, and $a = 0.8$, leading to an inappropriate exploration space; (d) DenseGRPO calibrates a timestep-specific noise intensity $\psi(t)$, enabling a suitable exploration space for all timesteps.

## 4 DENSEGRPO

In this section, we present DenseGRPO that aligns flow matching models using the step-wise dense rewards. Below, we begin by showing how to explicitly estimate the dense reward, evaluating the contribution of each step. Subsequently, we introduce the reward-aware scheme that calibrates the exploration space in the SDE sampler, providing a suitable exploration space for GRPO training.

### 4.1 STEP-WISE DENSE REWARD

As shown in Fig. 1, existing approaches estimate a single reward $R^i$ of the whole trajectory, and directly apply $R^i$ to optimize intermediate steps. Since $R^i$ is achieved by all steps, this manner encounters a mismatch between the trajectory-wise feedback signal and the step-wise contribution. To tackle this, we propose to estimate dense rewards that evaluate the contribution of each step, thereby providing a step-wise feedback signal. From the perspective of reward in RL, each action (*e.g.*, $\boldsymbol{x}_t^i$) receives a reward feedback (*e.g.*, $R_t^i$) that evaluates its corresponding future outcome. At timestep $= t$, the one-step denoising process $\boldsymbol{x}_t^i \rightarrow \boldsymbol{x}_{t-1}^i$ contributes to a reward raising from $R_t^i$ to $R_{t-1}^i$. Therefore, we define the step-wise dense reward $\Delta R_t^i$ of timestep $= \mathrm{t}$ as the reward gain:

$$\Delta R_t^i = R_{t-1}^i - R_t^i. \tag{7}$$

To this end, we first estimate the reward of any intermediate latent, *i.e.*, $R_t^i$. Typically, classical RL methods learn a critic function to immediately estimate the influence on the future outcome, which serves as a proxy of the reward at the current action (Pignatelli et al., 2023; Zhang et al., 2024). However, the critic function incurs increased training overhead and lacks adaptability to other tasks. In contrast, we implement a simple yet effective approach that eliminates the need for additional specialized models. Specifically, our approach leverages the deterministic nature of ODE sampler in flow matching models: given a latent $\boldsymbol{x}_t^i$ at timestep $= \mathrm{t}$, the ODE denoising trajectory, the corresponding clean latent, and hence the final clean image, are fully determined. Therefore, this one-to-one mapping allows the clean image obtained by ODE denoising to serve as a promising future counterpart for any latent $\boldsymbol{x}_t^i$. Building on these analyses, we propose that the reward of a latent $\boldsymbol{x}_t^i$ can be reliably assigned as that of the corresponding clean image via ODE denoising.

As illustrated in Fig. 2, for the $i$-th trajectory $\{x_t^i\}_{t=T}^0$ within a sampled group of GRPO, we first employ an $n$-step ODE denoising to obtain the underlying clean latent $\hat{x}_{t,0}^i$ for latent $x_t^i$:

$$\hat{x}_{t,0}^i = \text{ODE}_n(x_t^i, c). \tag{8}$$

Here, $\hat{x}_{0,0}^i = x_0^i$, and $\text{ODE}_n$ involves $n$ ODE denoising steps: $x_t^i \xrightarrow{\text{ODE}} ... \xrightarrow{\text{ODE}} \hat{x}_{t,\lfloor t/n \rfloor}^i \xrightarrow{\text{ODE}} \hat{x}_{t,0}^i$, where $\hat{x}_{t,\lfloor t/n \rfloor}^i$ is the latent generated by ODE sampler at $\text{timestep} = \lfloor t/n \rfloor$ and $n$ may be any integer in $[1, t]$. In our experiments, we set $n = t$ for improved performance (See Sec. 5.3 for its impact). After that, we decode the clean image from $\hat{x}_{t,0}^i$ and apply a reward model $\mathcal{R}$ to predict its reward $R_{t,0}^i$ as the latent reward for $x_t^i$:

$$R_t^i \triangleq R_{t,0}^i = \mathcal{R}(\hat{x}_{t,0}^i, c). \tag{9}$$

Notably, since $\hat{x}_{t,0}^i$ belongs to the clean distribution, plenty of established reward models can be seamlessly integrated as $\mathcal{R}$ for reward prediction. With the estimated $\{R_t^i\}_{t=1}^T$, we obtain the dense reward $\{\Delta R_t^i\}_{t=1}^T$ of $\text{timestep} = t$ by computing the reward gain via Eq. 7, which represents each step's contribution. During GRPO training, we replace the sparse $\mathcal{R}(x_0^i, c)$ in Eq. 2 with the dense $\Delta R_t^i$ at $\text{timestep} = t$, and thereby the advantage is calculated by:

$$\hat{A}_t^i = \frac{\Delta R_t^i - \text{mean}(\{\Delta R_t^i\}_{i=1}^G)}{\text{std}(\{\Delta R_t^i\}_{i=1}^G)}. \tag{10}$$

As a result, we align the reward signal with the contribution of denoising at each denoising step, facilitating effective policy optimization.

## 4.2 EXPLORATION SPACE CALIBRATION

Based on the estimated per-timestep dense reward above, a mismatch drawback between the exploration space and the denoising timestep schedule in existing GRPO-based methods is revealed. To promote diverse exploration for RL, Flow-GRPO proposes an SDE sampler that injects additional noise during trajectory sampling. This injection leads to a greater amount of noise than the denoising process, sampling out-of-distribution trajectories. Therefore, a suitable noise injection is critical, as an inappropriate setting often results in either excessive or insufficient stochasticity. However, the current uniform setting of noise injection fails to align with the time-varying nature of the

---

**Algorithm 1** Exploration Space Calibration

**Require:** policy model $p_\theta$, reward model $\mathcal{R}$, initial noise level $\psi(t)$, prompt dataset $\mathcal{C}$, total sampling steps $T$, number of samples $N$, small constants $\{\varepsilon_1, \varepsilon_2\}$
1: **for** iteration $k = 1, 2, ...$ **do**
2:     **for** sample $i = 1$ to $N$ **do**
3:         Init noise $x_T^i \sim \mathcal{N}(0, \mathbf{I})$
4:         Sample $c \sim \mathcal{C}$
5:         Sample a trajectory $\{x_t^i\}_{t=T}^0$ via SDE with $\psi(t)$
6:         Predict latent rewards $\{R_t^i = \mathcal{R}(\text{ODE}_n(x_t^i, c), c)\}_{t=T}^0$
7:         Calculate dense rewards $\{\Delta R_t^i = R_{t-1}^i - R_t^i\}_{t=T}^0$
8:     **end for**
9:     **for** timestep $t = T$ to 1 **do**
10:         **if** $|\text{num}(\{\Delta R_t^i > 0\}) - \text{num}(\{\Delta R_t^i < 0\})| < \varepsilon_1$ **then**
11:             $\psi(t) \leftarrow \psi(t) + \varepsilon_2$
12:         **else**
13:             $\psi(t) \leftarrow \psi(t) - \varepsilon_2$
14:         **end if**
15:     **end for**
16: **end for**
17: **return** $\psi(t)$

---

generation process, in which all timesteps share an identical noise level $a$ in Eq. 6. As plotted in Fig. 3 (a), we collect the step-wise dense reward of several trajectories with $a = 0.7$ using PickScore (Kirstain et al., 2023) as the reward model. The results show that all trajectories receive negative rewards at $\text{timestep} = 2$, indicating that nearly all samples in the current exploration space perform worse than the default. Lacking positive guidance, this inappropriate exploration space undermines effective policy optimization. We hypothesize that this issue may arise from the excessive noise injection in the current setting ($a = 0.7$). Hence, we further reduce the stochastic noise injection by lowering $a$ to 0.5. As depicted in Fig. 3 (b), this adjustment constrains the exploration space yet improves the reward balance, enabling a more fair distribution of positive and negative feedback, particularly at $\text{timestep} = 2$. Conversely, increasing the noise level to $a = 0.8$

Table 1: Performance on Compositional Image Generation, Visual Text Rendering, and Human Preference benchmarks, evaluated by task performance on test prompts, and by image quality and preference scores on DrawBench prompts. ImgRwd: ImageReward; UniRwd: UnifiedReward. UniRwd*: our evaluation results of the official checkpoints and our method with UnifiedReward. [1]

| Model | Task Metric | | | Image Quality | | Preference Score | | | |
|---|---|---|---|---|---|---|---|---|---|
| | GenEval↑ | OCR Acc.↑ | PickScore↑ | Aesthetic↑ | DeQA↑ | ImgRwd↑ | PickScore↑ | UniRwd↑ | UniRwd*↑ |
| SD3.5-M | 0.63 | 0.59 | 21.72 | 5.39 | 4.07 | 0.87 | 22.34 | 3.33 | 3.06 |
| *Compositional Image Generation* | | | | | | | | | |
| Flow-GRPO | 0.95 | — | — | 5.25 | 4.01 | 1.03 | 22.37 | 3.51 | 3.18 |
| Flow-GRPO+CoCA | 0.96 | — | — | 5.25 | 3.92 | 0.93 | 22.34 | - | 3.05 |
| DenseGRPO | 0.97 | — | — | 5.33 | 3.83 | 1.02 | 22.28 | - | 3.03 |
| *Visual Text Rendering* | | | | | | | | | |
| Flow-GRPO | — | 0.92 | — | 5.32 | 4.06 | 0.95 | 22.44 | 3.42 | 3.17 |
| Flow-GRPO+CoCA | — | 0.93 | — | 5.32 | 4.08 | 0.92 | 22.47 | - | 3.11 |
| DenseGRPO | — | 0.95 | — | 5.31 | 4.02 | 0.91 | 22.44 | - | 3.12 |
| *Human Preference Alignment* | | | | | | | | | |
| Flow-GRPO | — | — | 23.31 | 5.92 | 4.22 | 1.28 | 23.53 | 3.66 | 3.38 |
| Flow-GRPO+CoCA | — | — | 23.63 | 6.22 | 4.16 | 1.32 | 23.80 | - | 3.38 |
| DenseGRPO | — | — | 24.64 | 6.35 | 4.06 | 1.41 | 24.55 | - | 3.39 |

Figure 4: Comparison of learning curves. Figures (a) to (c) correspond to the tasks of compositional image generation, visual text rendering, and human preference alignment, respectively.

expands the exploration space, as evidenced by the greater diversity of rewards at $\text{timestep} = 10$ in Fig. 3 (c). However, a more pronounced imbalance arises at certain timesteps, *e.g.*, $\text{timestep} = 3$ and 2. These findings underscore the limitation that a uniform noise injection setting fails to produce a suitable exploration space for all timesteps. Therefore, a timestep-specific noise injection setting is expected to align with the time-varying nature of the generation process.

To mitigate this, we propose to calibrate the exploration space by adaptively adjusting the stochasticity injection in the SDE sampler suitable for all timesteps, yielding a timestep-specific noise level $\psi(t)$. We suggest that an ideal exploration space is supposed to provide diverse trajectories while preserving dense reward balance. Based on the above observation, a higher noise level facilitates exploration diversity, while a lower noise level benefits reward balance. Consequently, we advocate for using a higher feasible noise intensity to enhance exploration diversity, up to the point where reward imbalance occurs. As illustrated in Algorithm 1, we start by sampling plenty of trajectories $\{(\boldsymbol{x}_T^i, \boldsymbol{x}_{T-1}^i, ..., \boldsymbol{x}_0^i)\}_{i=1}^G$ and then predict their dense rewards $\{\Delta R_t^i\}$. Subsequently, for each timestep, we increase the noise level slightly when dense rewards are balanced (*i.e.*, the disparity between the number of positive and negative samples is minimal), or decrease otherwise. By iteratively updating, we obtain a suitable $\psi(t)$ output, which ensures a balanced exploration space for all timesteps. Accordingly, $\sigma_t$ in Eq. 6 is employed as $\sigma_t = \psi(t)\sqrt{\frac{t}{1-t}}$. Given that $\psi(t)$ is a self-adjusting function with respect to $t$, the item $\sqrt{\frac{t}{1-t}}$, which is constant for $t$, can be incorporated into the calibration process. Hence, we unify the formulation and employ $\sigma_t$ as follows:

$$\sigma_t = \psi(t). \tag{11}$$

---

[1] Our experiments reveal a discrepancy with the results reported in Flow-GRPO paper when evaluating the official checkpoints with UnifiedReward. This may stem from updates of the UnifiedReward checkpoint or the sglang package, as discussed in `https://github.com/yifan123/flow_grpo/issues/39`.

| SD 3.5-M | Flow-GRPO | Flow-GRPO+CoCA | DenseGRPO |
| --- | --- | --- | --- |

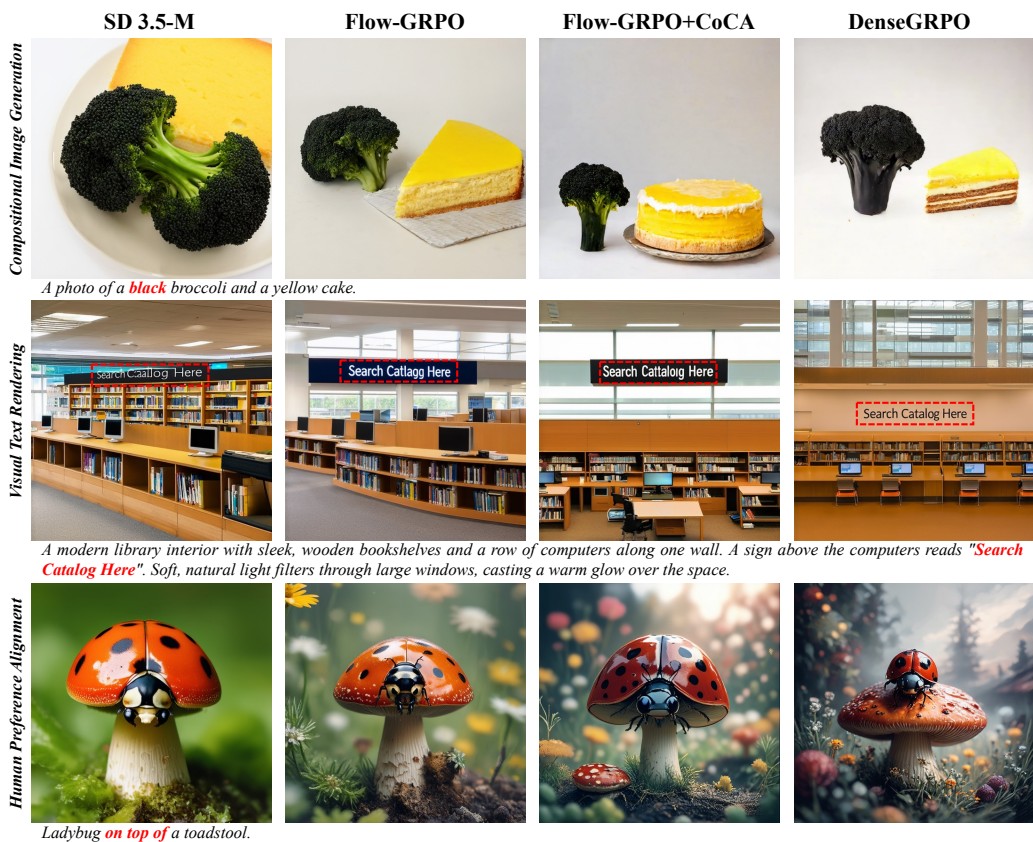

*A photo of a **black** broccoli and a yellow cake.*

*A modern library interior with sleek, wooden bookshelves and a row of computers along one wall. A sign above the computers reads **"Search Catalog Here"**. Soft, natural light filters through large windows, casting a warm glow over the space.*

*Ladybug **on top of** a toadstool.*

Figure 5: Qualitative comparison on three benchmarks: Compositional Image Generation, Visual Text Rendering, and Human Preference Alignment. Our DenseGRPO generates high-quality outcomes across all tasks, excelling in color accuracy, text fidelity, and content alignment.

## 5 EXPERIMENT

### 5.1 IMPLEMENTATION DETAIL

Following Flow-GRPO, we evaluate our method on three text-to-image tasks: (1) Compositional Image Generation, employing GenEval (Ghosh et al., 2023) as the reward model, (2) Human Preference Alignment, utilizing PickScore (Kirstain et al., 2023), (3) Visual Text Rendering, predicting OCR accuracy (Gong et al., 2025) as reward. The experimental setup aligns with Flow-GRPO, including a sampling timestep $T = 10$, an evaluation timestep $T = 40$, a group size $G = 24$, and an image resolution of 512. The KL ratio $\beta$ in Eq. 4 is set to 0.04 for compositional image generation and visual text rendering, and 0.01 for human preference alignment.

### 5.2 MAIN RESULT

We compare the proposed DenseGRPO with Flow-GRPO (Liu et al., 2025) and CoCA (Liao et al., 2025). Since the official CoCA is designed on DDPMs, we implement their core idea on flow matching models by tracking the latent similarity for step-wise reward, denoted by "Flow-GRPO+CoCA". As summarized in Tab. 1 and Fig. 4, our DenseGRPO achieves superior performance, outperforming competitors across all three tasks. Notably, in the task of human preference alignment, our DenseGRPO significantly surpasses the competitors by at least 1.01 of PickScore. In addition, compared to "Flow-GRPO+CoCA", which leverages latent similarity to estimate step-wise feedback signal, the substantial gains of our ODE-based approach validate its advancement to provide a more accurate dense reward. Moreover, as shown in Fig. 5, our DenseGRPO generates favorable outcomes with higher visual and semantic quality. For instance, in the third row, only our DenseGRPO successfully generates the positional relationship of "on top of", whereas other methods produce a combination of "ladybug" and "toadstool". These results demonstrate the significant advantages of DenseGRPO in aligning the target preference.

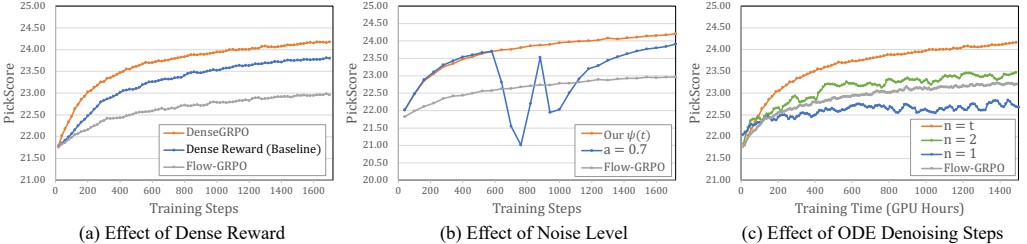

Figure 6: Ablation studies on our critical designs. (a) Step-wise dense reward aligns with contribution, surpassing trajectory-wise sparse reward. (b) Our time-specific noise level enables a suitable exploration space. (c) Increased ODE denoising steps ($n$) improve dense reward accuracy, yielding superior results. The vertical axis denotes the PickScore results. The horizontal axis of (a) and (b) is training steps, while the horizontal axis of (c) denotes training time for training cost comparison.

## 5.3 ANALYSIS

**Effect of Dense Reward.** To investigate whether RL benefits more from sparse rewards or dense rewards, we include another setting for comparison, namely "Dense Reward (Baseline)", which directly applies the $x_{t-1}^i$ reward ($R_{t-1}^i$) for optimizing denoising step at $\text{timestep} = t$. During GRPO training, its advantage is computed as $\hat{A}_t^i = \frac{R_{t-1}^i - \text{mean}(\{R_{t-1}^i\}_{i=1}^G)}{\text{std}(\{R_{t-1}^i\}_{i=1}^G)}$. As illustrated in Fig. 6 (a), "Dense Reward (Baseline)" offers greater benefits than Flow-GRPO, highlighting the effectiveness of dense reward. This advancement is further confirmed in Tab. 1 and Fig. 4. By employing step-wise rewards, "Flow-GRPO+CoCA" outperforms the vanilla Flow-GRPO. These findings highlight the critical role of step-wise dense rewards, which align the feedback signal more closely with the contribution for each denoising step, thereby facilitating policy optimization.

**Effect of Exploration Space Calibration.** To evaluate the effectiveness of the calibrated exploration space in Sec. 4.2, we make a comparison by applying the existing uniform setting ($a = 0.7$) in the proposed DenseGRPO. As presented in Fig. 6 (b), we find that our time-specific noise level advances the alignment task, indicating a more suitable exploration space for all timesteps and validating the success of our reward-aware calibration scheme. Besides, even if using the uniform $a = 0.7$ setting, our DenseGRPO also yields improved performance than Flow-GRPO, further validating DenseGRPO's superiority and the benefit of step-wise dense reward.

**Effect of Different ODE Denoising Steps.** The proposed DenseGRPO adopts an $n$-step ODE denoising (Eq. 8) to obtain clean latents. To evaluate the impact of different $n$, we perform ablation studies with $n = 1, 2$, and $t$, respectively. As depicted in Fig. 6 (c), we can draw two findings: (1) increasing the number of ODE denoising steps improves performance; (2) a single-step ODE yields suboptimal results, performing worse than Flow-GRPO. These findings suggest that a more accurate dense reward offers more benefits. Since existing reward models are primarily tailored for well-denoised images, utilizing more ODE steps is closer to a precise rollout, and thus receives more accurate rewards. In contrast, a single-step ODE deviates far from this domain, resulting in less accurate rewards and degraded performance. Furthermore, under the same experimental setting, $n = 1$, $n = 2$, and $n = t$ require 11, 13, and 19 GPU hours for training 20 steps, respectively. Although a larger $n$ incurs higher computational overheads, it offers improved performance with the same GPU training time, underscoring the critical role of dense reward accuracy.

**Discussion of Reward Hacking.** Following Flow-GRPO, we evaluate our method on Draw-Bench (Saharia et al., 2022) using four additional metrics: Aesthetic Score (Schuhmann, 2022), DeQA (You et al., 2025), ImageReward (Xu et al., 2023), and UnifiedReward (Wang et al., 2025c). As shown in Tab. 1, our DenseGRPO exhibits outstanding alignment capability with slight reward hacking in parts of tasks. Notably, in the human preference alignment, while achieving pronounced improvement on the PickScore metric, our method also performs strongly across other metrics. For example, in terms of the Aesthetic score, our DenseGRPO outperforms Flow-GRPO by 0.43, indicating more visually pleasant outcomes by DenseGRPO. These advancements demonstrate the strong robustness of the proposed DenseGRPO.

# 6 CONCLUSION

We present DenseGRPO to address the mismatch between trajectory-wise reward feedback and step-wise contribution. By estimating per-timestep dense rewards via an ODE-based approach, DenseGRPO aligns the reward feedback with the contribution of each denoising step, enabling a fine-grained credit assignment and facilitating effective optimization. Based on the estimated dense rewards, to address the current imbalance exploration in the SDE sampler, we propose a reward-aware scheme that calibrates timestep-specific noise injection, ensuring a suitable exploration space for all timesteps. Extensive experiments demonstrate the substantial gains achieved by the proposed DenseGRPO and validate the effectiveness of dense reward in flow matching model alignment.

## ETHICS STATEMENT

This work adheres to the ICLR Code of Ethics. All datasets utilized in this study are publicly available and used in accordance with their respective licenses. The research does not involve human subjects, sensitive personal information, or proprietary content. Besides, the methods proposed in this paper do not present any foreseeable risks of misuse or harm.

## REPRODUCIBILITY STATEMENT

We are committed to ensuring the reproducibility of our research. A comprehensive description of the proposed DenseGRPO is provided in Sec. 4. Implementation details, including the experimental setup, hyperparameter configurations, training pipeline, and evaluation metrics, are introduced in Sec. 5.1 and further elaborated in Sec. A of the Appendix. Additionally, all datasets utilized in this study are publicly available and described in detail in Sec. 5.1.

## ACKNOWLEDGMENT

This work is supported by the National Natural Science Foundation of China under grants U22B2053 and 623B2039.

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

## A    IMPLEMENTATION DETAIL

Our experiments are conducted based on the official implementation [1] of Flow-GRPO (Liu et al., 2025). The models are trained using 16 NVIDIA A100 GPUs. Before training, we first perform the exploration space calibration strategy to generate the noise level $\psi(t)$, as presented in Algorithm 1, where $\varepsilon_1$ and $\varepsilon_2$ are set to 2 and 0.01. Note that the obtained $\psi(t)$ is fixed in the training process. To ensure a fair comparison, we adopt the same experimental settings as Flow-GRPO. Specifically, we apply LoRA with $\alpha = 64$ and $r = 32$. During training, we use the AdamW optimizer with a learning rate of $3 \times 10^{-4}$, $\beta_1 = 0.9$, $\beta_2 = 0.999$, and a weight decay of $1 \times 10^{-4}$. The global batch size is set to 144, with a gradient accumulation step of 8. The total number of training iterations is 4500, 1500, and 4500 for the tasks of compositional image generation, visual text rendering, and human preference alignment tasks, respectively. Upon completing training, inference is conducted using the standard ODE sampler of flow matching models for text-to-image generation.

## B    MORE RESULT

### B.1    TRAINING CURVE OF KL LOSS

We present a visualization of the KL loss evolution during training in Fig. 7. The results show that the KL loss of Dense-GRPO is slightly larger than that of Flow-GRPO. This difference arises from the incorporation of the timestep-specific noise level, which encourages a more diverse exploration space and thereby pushes the model to deviate further from the original model.

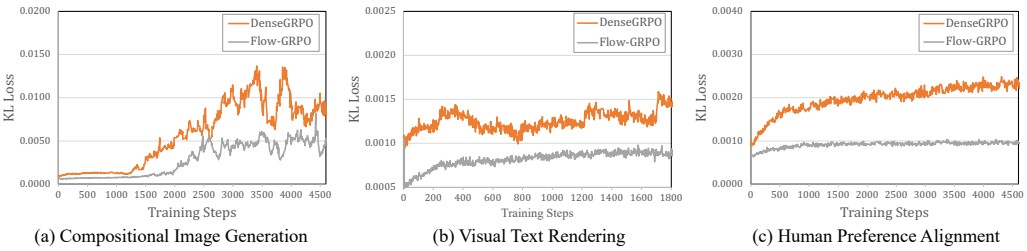

(a) Compositional Image Generation  (b) Visual Text Rendering  (c) Human Preference Alignment

Figure 7: Training curves of KL loss. Figures (a) to (c) correspond to the tasks of compositional image generation, visual text rendering, and human preference alignment, respectively.

### B.2    ACCURACY OF DENSEGRPO'S REWARD

In DenseGRPO, we estimate step-wise dense rewards by calculating the reward gain at each denoising step and utilize an ODE-based method to predict the reward for intermediate latents. To evaluate the accuracy of DenseGRPO's reward, we make a comparison between the predicted latent reward and the terminal reward of the SDE sampling trajectory with PickScore, as visualized in Fig. 8. Note that the latent reward at $\text{timestep} = 0$ is directly predicted by the reward model without requiring ODE sampling, and therefore corresponds to the terminal reward of the SDE sampling trajectory. The results show that the difference between the predicted latent rewards and the terminal trajectory rewards is minimal.

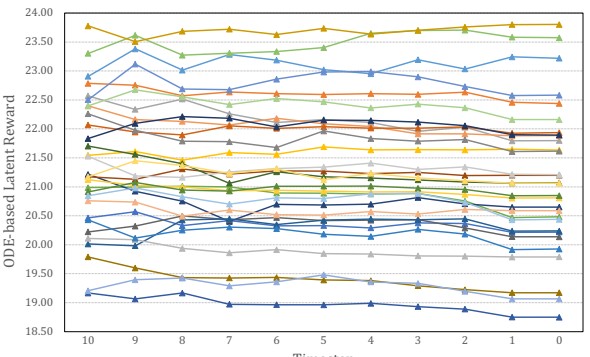

Figure 8: Visualization of ODE-based latent rewards, *i.e.*, $R_t^i$ predicted by Eq. 9, where each polyline denotes a sampled trajectory. $\text{timestep} = 0$ represents the terminal reward of the SDE sampling trajectory.

Furthermore, the relative ranking of rewards across different samples consistently aligns across all timesteps. These findings confirm the accuracy of the reward predictions in DenseGRPO.

---

[1] https://github.com/yifan123/flow_grpo

### B.3 MORE EXPERIMENT

**Experiment on FLUX.1-Dev.** We further evaluate the performance of our method against Flow-GRPO on FLUX.1-dev (Black et al., 2025) model using PickScore as the reward model. As shown in Fig. 9(a), the proposed DenseGRPO achieves substantial improvements over Flow-GRPO, suggesting the superiority and robustness of the estimated dense rewards.

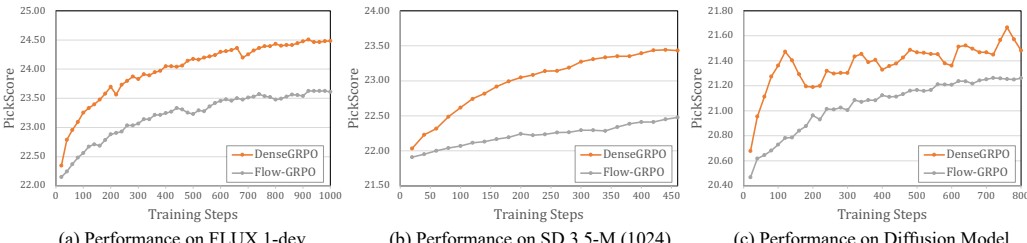

| (a) Performance on FLUX.1-dev | (b) Performance on SD 3.5-M (1024) | (c) Performance on Diffusion Model |

Figure 9: Performance of DenseGRPO compared with Flow-GRPO on additional models: (a) FLUX.1-dex, (b) SD 3.5-M on $1024 \times 1024$ resolution, and (c) diffusion model.

**Experiment on High Resolution.** As shown in Fig. 9(b), we raise the training and inference resolution to a higher resolution $1024 \times 1024$ on the SD 3.5-M model, utilizing PickScore as the reward model. The results reveal that DenseGRPO also yields a significant gain over Flow-GRPO, indicating the strong scalability of DenseGRPO.

**Experiment on Diffusion Model.** Though DenseGRPO focuses on flow matching models, it can also generalize to other generative models (Wei et al., 2024) by employing a deterministic sampler to predict dense rewards. This deterministic nature enables a one-to-one mapping between intermediate latents and clean latents, ensuring an accurate prediction of latent rewards and step-wise dense rewards. To validate this capability, we use SD 1.5 (Rombach et al., 2022) as the base model with an ODE sampler to predict $\hat{x}_{t,0}^i$ from $x_t^i$. The reward of $\hat{x}_{t,0}^i$ is then assigned to that of $x_t^i$, and the reward gain is calculated as the step-wise dense reward. As presented in Fig. 9(c), the performance improvement of dense reward within DenseGRPO demonstrates the accuracy and effectiveness of dense reward on diffusion models. These findings show that DenseGRPO is capable of generalizing to other generative families via a deterministic denoising sampler.

### B.4 REWARD HACKING ANALYSIS

Figure 10 illustrates examples of reward hacking. When GenEval is used as the reward model for compositional image generation, DenseGRPO achieves notable gains in compositional accuracy, such as object counting, but may occasionally experience a decline in image quality. A similar issue is observed in the task of visual text rendering. This problem arises from the step-wise dense reward in DenseGRPO, which aligns feedback with the contributions of individual steps, providing a more precise signal. While this increased reward accuracy enhances the learning process, it may also make the model more susceptible to overfitting the reward model, thereby amplifying the risk of reward hacking. One potential solution is to employ a large-scale reward model to provide higher-quality reward signals.

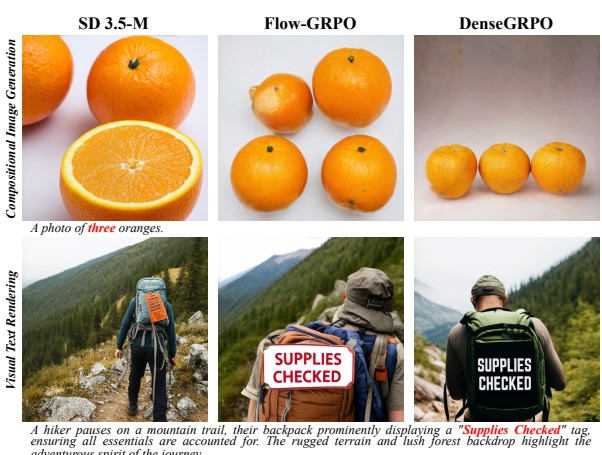

Figure 10: Visualization of reward hacking.

## C LLM USAGE

We use LLMs to assist with writing refinement, but do not involve them in core idea development.

