# OpenReview forum: "DenseGRPO: From Sparse to Dense Reward for Flow Matching Model Alignment"
_ICLR.cc/2026/Conference — ICLR 2026 Poster_

### Official Review · Reviewer_9Dzz · 2025-10-27

**Soundness:** 3
**Presentation:** 3
**Contribution:** 2
**Rating:** 4
**Confidence:** 3

**Summary:**

Recent GRPO-based flow matching models improve human preference alignment in text-to-image generation but still suffer from sparse rewards, where a single terminal reward is applied to all denoising steps. DenseGRPO addresses this by introducing dense, step-wise rewards that evaluate each denoising step via an ODE-based reward model. Furthermore, it adaptively calibrates the sampler’s exploration based on reward feedback, ensuring appropriate stochasticity across timesteps and yielding stronger alignment performance.

**Strengths:**

- This paper proposes a step-wise reward for RL finetuning.
- This paper designs an adaptive noise scale.

**Weaknesses:**

- There is no 'return' in Algorithm 1.
- There are many metrics in Table 1, and they have different directions for better performance. It will be clearer to add an arrow after them to denote which direction (up/down) is better.
- Only results of $a=0.7$  are provided in the ablation 'Effect of Exploration Space Calibration'. What about other $a$? Is $a=0.7$ the best hyperparameter?
- The return defined in reinforcement learning is $R_{t_0}=\sum_{t=t_0}^\infty \gamma^{t-t_0} r$, so the reward is not $R_{t+1}-R_t$.
- There is no code repository and reproducibility statement.

**Questions:**

- Why does the noise scale aim to guide the distribution of dense rewards to be balanced? Why not guide the distribution to fall in the positive reward region?
- In the ablation study 'Effect of Different ODE Denoising Steps', what about using $n=t/C$, where $C$ is a constant? This is proportional to $t$, while faster than $n=t$.

---

> ### Author Response · Authors · 2025-11-21
> **Response to Reviewer 9Dzz (Part 1)**
>
> We are grateful for your insightful comments and careful feedback on our paper. Below, we provide a detailed point-by-point response to each of your concerns.
>
> > **W1: There is no 'return' in Algorithm 1.**
>
> Thank you for your careful review. We have added "return" in the revision and have carefully reviewed the entire paper to ensure clarity and correctness.
>
>
> > **W2: There are many metrics in Table 1, and they have different directions for better performance. It will be clearer to add an arrow after them to denote which direction (up/down) is better.**
>
> All the metrics utilized in our study have been defined such that higher values indicate better performance. We have added directional arrows following each metric to clarify this. We sincerely appreciate your feedback on improving the presentation of our work.
>
>
> > **W3: Only results of $a=0.7$ are provided in the ablation 'Effect of Exploration Space Calibration'. What about other $a$? Is $a=0.7$ the best hyperparameter?**
>
> Thank you for your detailed comment. Higher $a$ boosts image diversity and exploration, yet injecting too much noise degrades image quality. Flow-GRPO has studied the effect of $a$ and demonstrated that $a=0.7$ is the best choice, which strikes a balance between exploration and stable training. Therefore, we employ the $a=0.7$ for comparison.
>
>
> > **W4: The return defined in reinforcement learning is $R_{t_0}=\sum_{t=t_0}^{\infty}\gamma^{t-t_0} r$, so the reward is not $R_{t+1}-R_t$.**
>
> Thank you for your careful review. We found a misunderstanding may exist between the reward $r$ and our reward $\Delta R^i_t=R^i_{t-1}-R^i_t$ (Eq. 7). Specifically, the mentioned $R_{t_0}$ considers the action rewards $r$ along a **single trajectory**. However, as described in Fig. 2, our reward $\Delta R^i_t=R^i_{t-1}-R^i_t$ involves **two trajectories** starting from $x\_t^{i}$:
>
> (1) `trajectory_t`: which applys $t$ ODE denoising steps to obtain $\hat{x}\_{t,0}^i$ and predicts its reward as $R\_t^i$;
>
> (2) `trajectory_(t-1)`: which first performs one SDE denoising step to get $x\_{t-1}^i$ and then adopts $t-1$ ODE denoising steps to obtain $\hat{x}\_{t-1,0}^i$, predicting its reward as $R\_{t-1}^i$.
>
> Therefore, $R^i_{t-1}$ and $R^i_t $ are predicted by different trajectories, and the dense reward $\Delta R^i\_t=R^i\_{t-1}-R^i\_t$ does not correspond to the reward $r$ you mentioned.
>
> Our motivation for defining this reward is that the deterministic nature of ODE sampling inherently lacks exploratory behavior. Once the current state is established, all subsequent ODE steps are fully defined and deterministic. As such, we can choose to assign a reward of zero for the ODE denoising step. Since the actions before $\rm{timestep=t}$ are identical for both trajectories, the $\Delta R^i\_t=R^i\_{t-1}-R^i\_t$ represents the reward attributed specifically to the SDE denoising step in `trajectory_(t-1)`, i.e., the denosing step at $\rm{timestep=t}$.
>
> We hope this clarification resolves any ambiguity and offers a clearer understanding of our reward formulation.
>
> > **W5: There is no code repository and reproducibility statement.**
>
> We are committed to ensuring the reproducibility of our research, and we have supplemented the reproducibility statement in the main paper. Specifically, a comprehensive description of the proposed DenseGRPO is provided in Sec. 4. Implementation details, including the experimental setup, hyperparameter configurations, training pipeline, and evaluation metrics, are introduced in Sec. 5.1 and further elaborated in Sec. A of the Appendix. Additionally, all datasets utilized in this study are publicly available and described in detail in Sec. 5.1.
>
> Besides, we have incorporated an ethics statement and the usage of LLMs in the revision. We sincerely appreciate your helpful recommendation, which has largely contributed to improving the thoroughness and quality of our paper.

---

> ### Author Response · Authors · 2025-11-21
> **Response to Reviewer 9Dzz (Part 2)**
>
> > **Q1: Why does the noise scale aim to guide the distribution of dense rewards to be balanced? Why not guide the distribution to fall in the positive reward region?**
>
> Thank you for your insightful comment. As mentioned in the Sec. 4.2 of the main paper, in flow matching models, the noise amount of the entire denoising process is generally considered as a constant. To promote diverse exploration for RL, an SDE sampler injects additional noise during training. However, this injection leads to a greater amount of noise than the denoising process, causing the denoised results produced by the SDE sampler to remain noisy and out-of-distribution. As a result, negative dense rewards persist and often constitute the majority, particularly when a higher level of noise injection (i.e, $a$) is supposed to be adopted for diverse exploration. Thus, it is rarely observed that the distribution of dense rewards falls in the positive reward region, particularly when adopting a high noise level. Therefore, we calibrate the noise level to obtain a balanced reward distribution, facilitating exploration diversity.
>
>
> > **Q2: In the ablation study 'Effect of Different ODE Denoising Steps', what about using $n=t/C$, where $C$ is a constant? This is proportional to $t$, while faster than $n=t$.**
>
> Thank you for your constructive suggestion. We conduct experiments with PickScore using $n=t/C$, where $C$ is set to 3 and 2, respectively. As shown below, we observed that in the early training stage (200 GPU Hours), $n=t/2$ provides better results compared to $n=t$. However, in the later training stage, $n=t$ outperforms both $n=t/2$ and $n=t/3$ under the same training duration. The underlying reason is that a small $n$ accelerates the training process but reduces the accuracy of reward predictions, thereby affecting the performance ceiling. Consequently, the smaller $n$ fails to achieve the performance of $n=t$ during extended training. As a result, we recommend the setting $n=t$ for better performance.
>
>
> | Training Time (GPU Hours) |  200  |  400  |  600  |  800  |
> | ---- | :----: | :----: | :----: | :----: |
> | Flow-GRPO                 | 22.55 | 22.78 | 22.94 | 23.01 |
> | DenseGRPO ($n=t/3$)        | 23.13 | 23.48 | 23.58 | 23.77 |
> | DenseGRPO ($n=t/2$)        | **23.19** | 23.52 | 23.62 | 23.82 |
> | DenseGRPO ($n=t$)          | 23.05 | **23.55** | **23.74** | **23.88** |

---

> > ### Comment · Reviewer_9Dzz · 2025-11-25
> >
> > Thanks for the detailed responses from the authors. Most of my concerns have been addressed, so I raise my score to 6.

---

> > > ### Author Response · Authors · 2025-11-27
> > > **Thank you for your recognition and score update**
> > >
> > > Thank you very much for your kind decision to raise the score! We are grateful for your recognition of our rebuttal and your positive evaluation. We sincerely appreciate the valuable and careful feedback you provided throughout this process, which has been instrumental in improving this manuscript.

---

### Official Review · Reviewer_Fc1V · 2025-10-28

**Soundness:** 3
**Presentation:** 2
**Contribution:** 3
**Rating:** 6
**Confidence:** 3

**Summary:**

This paper introduces DenseGRPO, a novel reinforcement learning framework for aligning flow matching models with human preferences in text-to-image generation. The key innovation addresses the "sparse reward problem" in existing GRPO-based approaches, where a single terminal reward for the entire denoising trajectory is naively applied to optimize all intermediate steps, creating a mismatch between global feedback and fine-grained step-wise contributions.

DenseGRPO proposes two main components: (1) **Step-wise Dense Reward Estimation**: Instead of using a single terminal reward, the method predicts reward gains at each denoising step by applying a reward model to intermediate clean images obtained through ODE-based denoising. This approach assigns $\Delta R_t^i = R_{t-1}^i - R_t^i$ as the dense reward for each step, aligning feedback signals with actual step contributions. (2) **Reward-Aware Exploration Space Calibration**: Based on the estimated dense rewards, the authors identify that existing uniform noise injection in SDE samplers creates inappropriate exploration spaces. They propose an adaptive scheme that adjusts timestep-specific stochasticity  to balance dense reward distributions across all timesteps.

Experiments on three benchmarks (compositional image generation, visual text rendering, and human preference alignment) demonstrate superior performance over Flow-GRPO and CoCA baselines. The method achieves notable improvements, particularly +1.01 PickScore on human preference alignment, while maintaining good generalization across multiple evaluation metrics.

**Strengths:**

**Originality:** The paper presents a genuinely novel perspective on the sparse reward problem in flow matching model alignment. While dense rewards have been explored in text generation and diffusion models, the specific ODE-based approach for estimating step-wise rewards without additional specialized models is creative and practical. The identification of exploration space mismatch through dense reward analysis is an insightful observation that leads naturally to the second contribution. The paper distinguishes itself from CoCA by actually training with step-wise rewards rather than just using trajectory-wise signals proportionally weighted.

**Quality:** The experimental validation is comprehensive, covering three distinct tasks with multiple evaluation metrics. The ablation studies systematically validate each design choice: dense vs. sparse rewards (Figure 6a), exploration space calibration (Figure 6b), and ODE denoising steps (Figure 6c). The visualization of dense reward distributions (Figure 3) provides compelling evidence for the exploration space problem. The method demonstrates consistent improvements across all tasks, with particularly strong results on human preference alignment. The authors appropriately acknowledge discrepancies in reproducing baseline results with UnifiedReward and provide their own evaluations for fair comparison.

**Clarity:** The paper is generally well-written with clear motivation. Figure 1 effectively illustrates the core problem and proposed solution. The mathematical formulation is precise, and the algorithm pseudocode (Algorithm 1) provides implementation details. The progression from problem identification to solution is logical and easy to follow.

**Significance:** The work addresses a fundamental limitation in current GRPO-based alignment methods for flow matching models. The ODE-based dense reward estimation is elegant and can integrate with any existing reward model, making it broadly applicable. The reward-aware exploration calibration provides insights that could benefit other RL-based generative model training approaches. The consistent improvements across diverse tasks suggest practical value for the community.

**Weaknesses:**

**1. Limited Analysis of Reward Hacking:** While Table 1 shows some reward hacking (e.g., Aesthetic Score degradation in compositional generation), the discussion is superficial. The paper doesn't analyze: (a) why certain metrics degrade while others improve, (b) whether dense rewards exacerbate or mitigate reward hacking compared to sparse rewards, (c) qualitative failure cases where the model exploits the reward function, or (d) potential solutions. Given that reward hacking is a critical concern in RL-based alignment, this deserves deeper investigation.

**2. Generalization Questions:** Several aspects of generalization remain unexplored: (a) Does the method work with different base models beyond SD3.5-M? (b) How does it scale to higher resolution generation or longer sampling trajectories? (c) Does it apply to other generative models (e.g., diffusion models, consistency models)? (d) How does it perform on out-of-distribution prompts or edge cases?

**3. Writing and Presentation Issues:** Minor clarity issues: (a) Notation inconsistency: sometimes $x_t^i$ and sometimes $x^i_t$, (b) Figure 2 could be clearer about which components run during training vs. inference, (c) The connection between ODE denoising steps n and timestep t is confusing ("n may be any integer in [1,t]" but then "we set n=t"), (d) Some experimental details are missing (e.g., learning rates, number of training iterations to convergence, batch sizes).

**Questions:**

1. How many iterations does Algorithm 1 require for $\psi(t)$ to converge? What are the sensitivity and recommended values for hyperparameters $\epsilon_1$ and $\epsilon_2$?
2. Is exploration space calibration performed once as preprocessing or continuously during training? If continuous, how frequently?
3. How does the accuracy of dense reward estimates degrade as you move further back in timesteps (larger t)? Have you analyzed the reward estimation error at different timesteps?

---

> ### Author Response · Authors · 2025-11-21
> **Response to Reviewer Fc1V (Part 1)**
>
> Thank you very much for your thoughtful review of our manuscript and for your valuable suggestions. We provide a point-by-point response to your comments below.
>
> > **W1: Limited Analysis of Reward Hacking: While Table 1 shows some reward hacking (e.g., Aesthetic Score degradation in compositional generation), the discussion is superficial. The paper doesn't analyze: (a) why certain metrics degrade while others improve, (b) whether dense rewards exacerbate or mitigate reward hacking compared to sparse rewards, (c) qualitative failure cases where the model exploits the reward function, or (d) potential solutions. Given that reward hacking is a critical concern in RL-based alignment, this deserves deeper investigation.**
>
> Thank you for your valuable comment. We have briefly discussed reward hacking in Paragraph 4 of Sec. 5.3, suggesting that our DenseGRPO exhibits slight reward hacking in parts of tasks. We provide qualitative failure cases in Sec. B.4 of the Appendix. When GenEval is used as the reward model for compositional image generation, DenseGRPO achieves notable gains in compositional accuracy, such as object counting, but may occasionally experience a decline in image quality. A similar issue is observed in the task of visual text rendering. This issue arises from step-wise dense rewards within DenseGRPO, which align feedback signals with the contributions of individual steps, providing a more precise reward signal. While this increased reward accuracy enhances the learning process, it can also make the model more susceptible to overfitting the reward model, thereby increasing the risk of reward hacking. As a result, though significant gains are achieved in the training reward, performance on other evaluation metrics may experience slight degradation. A potential solution could involve leveraging a large-scale reward model to produce higher-quality reward signals, as such models maintain high reward variance during RL fine-tuning, demonstrating resistance to reward hacking and enabling the generation of diverse, high-quality outputs[1].
>
> [1] Jie Wu, et al. Rewarddance: Reward scaling in visual generation. arXiv preprint, 2025.

---

> ### Author Response · Authors · 2025-11-21
> **Response to Reviewer Fc1V (Part 2)**
>
> > **W2: Generalization Questions: Several aspects of generalization remain unexplored: (a) Does the method work with different base models beyond SD3.5-M? (b) How does it scale to higher resolution generation or longer sampling trajectories? (c) Does it apply to other generative models (e.g., diffusion models, consistency models)? (d) How does it perform on out-of-distribution prompts or edge cases?**
>
> Below, we respond to each of your questions about generalization one by one.
>
> (a) **Different base models.** We have applied DenseGRPO on FLUX.1-Dev[1] model and reported the results in the Sec. B.3 of the Appendix. The substantial improvements of DenseGRPO over Flow-GRPO demonstrate the superiority and robustness of the estimated dense rewards on different models.
>
> (b) **Higher resolution generation or longer sampling trajectories.** We raise the training and inference resolution to a higher resolution $1024\times1024$ on the SD 3.5-M model, utilizing PickScore as the reward model. As presented in the Sec. B.3 of the Appendix, DenseGRPO also yields a significant gain over Flow-GRPO, indicating its strong scalability. Regarding the longer sampling trajectories, Flow-GRPO has shown that utilizing 10 training steps is a suitable choice in GRPO training for flow matching models, as it accelerates convergence while achieving comparable performance to that of longer sampling trajectories. When evaluated with 40 steps, the models (both Flow-GRPO and DenseGRPO) generate visually pleasing outputs, as presented in Fig. 5.
>
> (c) **Other generative models.** We perform an experiment on the diffusion model, as shown in Sec. B.3 of the Appendix. Specifically, we train on SD 1.5[2] model with an ODE sampler to predict $\hat{x}^{i}\_{t,0}$ from the $x^i\_t$. The reward of $\hat{x}^{i}\_{t,0}$ is then assigned to that of $x^i\_t$, and the reward gain is calculated as the dense reward. As shown in Fig. 9(c), the dense reward of DenseGRPO surpasses the sparse reward within Flow-GRPO, thereby demonstrating the accuracy and effectiveness of dense reward on diffusion models. These findings show that DenseGRPO is capable of generalizing to other generative models.
>
> (d) **Out-of-distribution evaluation.** In Tab. 1, we have provided an evaluation on the out-of-distribution prompts (DrawBench prompts) and present these results below, which were trained with PickScore as the reward model, based on the prompts provided by Flow-GRPO. The results demonstrate a remarkable improvement of 1.02 PickScore on the out-of-distribution prompts, suggesting an outstanding generalization ability.
>
> | Model     | PickScore (DrawBench Prompts) |
> | --------  | :--------: |
> | SD3.5-M   | 22.34    |
> | Flow-GRPO | 23.53    |
> | **DenseGRPO** | **24.55** |
>
> Furthermore, following Flow-GRPO, we train SD 3.5-M model on 60 object classes and evaluate on 20 unseen classes in the task of compositional image generation. As the results presented below, DenseGRPO yields superior performance across all evaluated metrics, including object generation, color accuracy, and spatial relations, demonstrating its robust generalization capability in unseen scenarios.
>
> | Model (1200 Training Steps) | Overall | Single Obj. | Two Obj. | Counting | Colors | Position | Attr. Binding |
> | -------- | :--------: | :--------: | :--------: | :--------: | :--------: | :--------: | :--------: |
> |SD 3.5-M| 0.64 | 0.96 | 0.73 | 0.53 | 0.87 | 0.26 | 0.47 |
> |Flow-GRPO| 0.76 | 0.97 | 0.85 | 0.77 | 0.91 | 0.46 | 0.71 |
> |**DenseGRPO**| **0.82** | **1.00** | **0.88** | **0.83** | **0.94** | **0.55** | **0.82** |
>
>
> [1] Forest Labs Black, et al. Flux.1 kontext: Flow matching for in-context image generation and editing in latent space. arXiv preprint, 2025.
>
> [2] Robin Rombach, et al. High-resolution image synthesis with latent diffusion models. CVPR, 2022.

---

> ### Author Response · Authors · 2025-11-21
> **Response to Reviewer Fc1V (Part 3)**
>
> > **W3: Writing and Presentation Issues: Minor clarity issues: (a) Notation inconsistency: sometimes $x_t^i$ and sometimes $x_t^i$, (b) Figure 2 could be clearer about which components run during training vs. inference, (c) The connection between ODE denoising steps n and timestep t is confusing ("n may be any integer in [1,t]" but then "we set n=t"), (d) Some experimental details are missing (e.g., learning rates, number of training iterations to convergence, batch sizes).**
>
> We sincerely apologize for these presentation issues and have carefully corrected these issues for clarity improvement, as detailed below.
>
> (a) **Notation inconsistency.** We found that this inconsistency mainly appears in Paragraph 2 and Paragraph 3 of Sec. 4.1, which has been corrected in the revision.
>
> (b) **Training v.s. Inference illustration.** Aligning with common practice in this area, the inference process is conducted using the standard ODE sampler of flow matching models. Since DenseGRPO focuses on the training process, we have retained the current version of Fig. 2; for clarity, we have supplemented the inference details in the implementation section (see Sec. A of the Appendix). We greatly appreciate your understanding and remain open to any further concerns or suggestions you may have.
>
> (c) **Confusion between $n$ and $t$.** In the proposed DenseGRPO, the ODE denoising steps $n$ is a hyperparameter, which can be set as any integer in [1,t]. A large $n$ yields better performance but requires additional computational overhead, as studied in Sec. 5.2 and Fig. 6(c). In our experiments, we adopt the choice $n=t$ for improved performance, and the effect of $n$ has already been studied in Sec. 5.3.
>
> (d) **Additional experimental details.** We have supplemented the experimental details in Sec. A of the Appendix. For your convenience, these details are also presented below.
>
> _"Our experiments are conducted based on the official implementation (https://github.com/yifan123/flow_grpo) of Flow-GRPO. The models are trained using 16 NVIDIA A100 GPUs. Before training, we first perform the exploration space calibration strategy to generate the noise level $\psi(t)$, as presented in Algorithm 1, where $\varepsilon_1$ and $\varepsilon_2$ are set to 2 and $0.01$. Note that the obtained $\psi(t)$ is fixed in the training process. To ensure a fair comparison, we adopt the same experimental settings as Flow-GRPO. Specifically, we apply LoRA with $\alpha = 64$ and $r = 32$. During training, we use the AdamW optimizer with a learning rate of $3\times10^{-4}$, $\beta_1 = 0.9$, $\beta_2 = 0.999$, and a weight decay of $1\times10^{-4}$. The global batch size is set to 144, with a gradient accumulation step of 8. The total number of training iterations is 4500, 1500, and 4500 for the tasks of compositional image generation, visual text rendering, and human preference alignment tasks, respectively. Upon completing training, inference is conducted using the standard ODE sampler of flow matching models for text-to-image generation."_
>
> Additionally, we have conducted a thorough review of the entire paper to ensure the accuracy and clarity of the expression.

---

> ### Author Response · Authors · 2025-11-21
> **Response to Reviewer Fc1V (Part 4)**
>
> > **Q1: How many iterations does Algorithm 1 require for $\psi (t)$ to converge? What are the sensitivity and recommended values for hyperparameters $\varepsilon_1$ and $\varepsilon_2$ ?**
>
> Thank you for your detailed review. Algorithm 1 requires 50 iterations to converge for all tasks, and we set $\varepsilon_1=2$ and $\varepsilon_2=0.01$ based on two considerations.
> First, the $\varepsilon_1 \sim \lbrace1,2,3,...\rbrace$ controls the difference in counts between positive and negative samples, with a smaller $\varepsilon_1$ yielding a more balanced exploration space. Ideally, $\varepsilon_1=1$ would be the optimal choice, as it ensures the smallest possible difference. However, achieving an equal number of positive and negative samples in each instance is infeasible in practice. Consequently, we have relaxed the constraint and adopted the next optimal value of $\varepsilon_1=2$.
> Second, a smaller value of $\varepsilon_2$ enables finer adjustments, similar to the role of a learning rate in machine learning algorithms. As the noise level is typically on the precision scale of 0.1, setting a smaller magnitude $\varepsilon_2=0.01$ allows for more fine-grained adjustments.
> Further reducing the magnitude, for example $\varepsilon_2=0.001$, leads to slower convergence and increases the required number of iterations for convergence (50 iterations v.s. 800 iterations). In the revision, we supplement these hyperparameter settings in Sec. A of the Appendix.
>
>
> > **Q2: Is exploration space calibration performed once as preprocessing or continuously during training? If continuous, how frequently?**
>
> We perform the exploration space calibration during preprocessing before training and fix the results during training. For clarity, we add this detail in Sec. A in the Appendix.
>
>
> > **Q3: How does the accuracy of dense reward estimates degrade as you move further back in timesteps (larger t)? Have you analyzed the reward estimation error at different timesteps?**
>
> Thank you for your insightful suggestion. DenseGRPO estimates the step-wise dense reward by computing the gain of the latent reward at each timestep. Therefore, the accuracy of step-wise dense reward relies on the accuracy of latent reward, which is predicted by an ODE-based approach. To evaluate this, we compare the predicted latent rewards with the terminal rewards in Sec. B.2 of the Appendix. While existing methods[1] fail to accurately estimate rewards in high-noise steps (larger t), the results in Figure 8 illustrate that the relative ordering of rewards among different samples remains stable across all timesteps. Therefore, the discrepancy between the predicted latent rewards and the terminal trajectory rewards is negligible across all steps, including high-noise steps, demonstrating the robustness and precision of the DenseGRPO's reward.
>
> [1] Jiazheng Xu, et al. Imagereward: Learning and evaluating human preferences for text-to-image generation. NIPS, 2023.

---

> ### Comment · Reviewer_Fc1V · 2025-11-22
>
> I appreciate the author's reply and additional experiments; I found them all very useful and helpful, resolving many of my previous concerns. I appreciate the additional experiments in the appendix in more base models, which I believe greatly enhance the convince of the proposed method. With that in mind, especially with the improvement of some presenation/writing issues, I find the current paper  easier to follow, so I plan to raise my score for Presentation to 3. Furthermore, I believe that current papers can benefit from addressing or discussing some of the following concerns and minor issues:
>
> (1) If the difference between $\boldsymbol{x}$ and $\hat{\boldsymbol{x}}$ could be explained more clearly in section 4.1, the article would be easier to understand and clearer.
>
> (2) The font size of the legend in Figure 6 appears to be a bit small, which may cause misunderstanding (e.g., t and 1) or reduce the effectiveness of the image.
>
> (3) Could you elaborate on the differences between the proposed dense reward and CoCA? And why does the proposed dense reward achieve better results compared to CoCA+Flow-GRPO?
>
> (4) The paper compares Flow-GPRO and Flow-GRPO+CoCA. I'm curious if the authors could further elaborate on the differences and advantages between the proposed method and the critic-based methods, such as better performance, which I believe, could enhance the paper's significance.
>
> (5) Regarding the n-step ODE denoising, is the performance best when n=t? Has it been explored other setting of n can further improve performance or provide more accurate intermediate state rewards?
>
> (6) The paper extensively mentions the results under different settings of noise level control 'a'. Since these results may be far  from the definition of 'a', briefly mentioning the meaning of 'a' in the caption could avoid confusion.

---

> > ### Author Response · Authors · 2025-11-27
> > **Further Response to Reviewer Fc1V (Part 1)**
> >
> > Thank you for your thoughtful and timely feedback. We are encouraged to know that our responses have successfully addressed your concerns and that you have decided to raise your evaluation of our work. To further address your additional concerns, we provide a detailed point-by-point response below. We notice that both **Question 3** and **Question 4** pertain to the CoCA method. Therefore, we provide a comprehensive and detailed combined response to address the two questions together.
> >
> >
> > > **Q1: If the difference between $x$ and $\hat{x}$ could be explained more clearly in section 4.1, the article would be easier to understand and clearer.**
> >
> > In our paper, ${x}^{i}_t$ denotes the latent generated by the SDE sampler, while $\hat{x}^{i}_t$ represents the latent generated by the ODE sampler. We have clarified this point in the revision (see Paragraph 3, Sec. 4.1).
> >
> >
> > > **Q2: The font size of the legend in Figure 6 appears to be a bit small, which may cause misunderstanding (e.g., t and 1) or reduce the effectiveness of the image.**
> >
> > Thank you for your detailed review. We have revised Fig. 6 to enhance its presentation quality, including increasing the font size. Additionally, we have carefully reviewed all figures throughout the paper to ensure clarity and readability.

---

> > ### Author Response · Authors · 2025-11-27
> > **Further Response to Reviewer Fc1V (Part 2)**
> >
> > > **Q3: Could you elaborate on the differences between the proposed dense reward and CoCA? And why does the proposed dense reward achieve better results compared to CoCA+Flow-GRPO? & Q4: The paper compares Flow-GPRO and Flow-GRPO+CoCA. I'm curious if the authors could further elaborate on the differences and advantages between the proposed method and the critic-based methods, such as better performance, which I believe, could enhance the paper's significance.**
> >
> > Thank you for your insightful review. We would like to address your concerns from the following aspects.
> >
> > (1) **Flow-GRPO+CoCA Detail.** As mentioned in our paper, the official CoCA[1] is designed for DDPMs and tracks the latent similarity for **latent reward**. For a fair comparison, we implement their core idea on flow matching models and build the baseline "Flow-GRPO+CoCA". Specifically, given a $T$-timestep denoising trajectory $(x^i\_T, x^i\_{T-1}, ...,x^i\_{t},..., x^i\_0)$ and a reward $R^i$ of the image decoded from $x^i\_0$. Flow-GRPO+CoCA first computes the latent change by calculating the cosine similarity $\lbrace\omega\_t^i\rbrace\_{t=1}^T$ between latent at each denoising step and clean latent: $\omega_t^i = \mathrm{CosSim}(x^i\_t, x^i\_0)$, where $\mathrm{CosSim}(·,·)$ denotes the cosine similarity calucation. Based on $\lbrace\omega\_t^i\rbrace\_{t=1}^T$, each latent's reward $R_t^i$ is estimated as $R_t^i = \omega\_t^i R^i$. To establish a baseline method for comparison, **we utilize the estimated latent reward of CoCA to compute the dense reward proposed by DenseGRPO**, which assigns the reward gain of each denoising step as the step-wise reward $\Delta R_t^i=R_{t-1}^i-R_t^i$ for GRPO training.
> >
> >
> > (2) **Flow-GRPO v.s. Flow-GRPO+CoCA.** The primary difference between Flow-GRPO and Flow-GRPO+CoCA lies in the reward assignment strategy for each denoising step. As summarized below, Flow-GRPO utilizes a trajectory-wise, sparse reward $R^i$ for each step, whereas Flow-GRPO+CoCA assigns a step-wise, dense reward $\Delta R_t^i=R_{t-1}^i-R_t^i$ to each denoising step. Therefore, compared with Flow-GRPO, Flow-GRPO+CoCA adopts a more precise credit assignment, alleviating the mismatch issue between the global feedback signals and the fine-grained contributions at the intermediate denoising step within Flow-GRPO. Consequently, as shown in Tab. 1 and Fig. 4, Flow-GRPO+CoCA achieves better performance than Flow-GRPO, highlighting the critical role of the dense rewards in flow matching model alignment.
> >
> >
> > (3) **DenseGRPO v.s. Flow-GRPO+CoCA.** Flow-GRPO+CoCA and DenseGRPO share a common approach of leveraging step-wise dense reward prediction through credit assignment for GRPO training. This approach predicts rewards for intermediate latent variables and assigns the corresponding reward gains as dense rewards for each step. The key distinction between the two methods lies in their approach to predicting latent rewards. Specifically, Flow-GRPO+CoCA allocates the terminal reward to each latent based on latent similarity, which suffers from two limitations. First, while latent changes indicate the influence on the outcome, they do not effectively capture the quality of the final result. Second, the difference between the latent feature space and the reward model's feature space prevents changes in latent features from accurately reflecting variations in rewards. As a result, although similarity-based allocation allows for the formulation of dense rewards, the accuracy of this method remains limited. To achieve more accurate dense rewards, our DenseGRPO introduces an ODE-based approach that independently predicts the reward for each latent. The deterministic nature of this ODE-based method allows for precise prediction of latent rewards, significantly enhancing the accuracy of step-wise dense rewards. As demonstrated, DenseGRPO outperforms Flow-GRPO+CoCA in its ability to align models with human preferences.
> >
> >
> >
> > |  | Flow-GRPO | Flow-GRPO+CoCA | DenseGRPO |
> > | -------- | -------- | -------- | -------- |
> > | Assigned Step Reward | Trajectory-wise Sparse Reward $R^i$ | Our Step-wise Dense Reward $\Delta R_t^i=R_{t-1}^i-R_t^i $ | Our Step-wise Dense Reward $\Delta R_t^i=R_{t-1}^i-R_t^i $|
> > | Predicted Latent Reward| - | $R_t^i = \mathrm{CosSim}(x^i_t, x^i_0) R^i$ | $R_t^i=\mathcal{R}(\mathrm{ODE}_n(x^{i}_t, c),c)$ |
> >
> >
> >
> > [1] Xinyao Liao, et al. Step-level Reward for Free in RL-based T2I Diffusion Model Fine-tuning. arXiv preprint, 2025.

---

> ### Author Response · Authors · 2025-11-27
> **Further Response to Reviewer Fc1V (Part 3)**
>
> > **Q5: Regarding the n-step ODE denoising, is the performance best when n=t? Has it been explored other setting of n can further improve performance or provide more accurate intermediate state rewards?**
>
> Yes, our experiments reveal that $n=t$ yields the best performance. The ablation study of $n$ has already been presented in Paragraph 3, Sec. 5.3. From this study, we derive two key observations: (1) increasing the number of ODE denoising steps enhances performance, and (2) DenseGRPO with a single-step ODE performs worse than Flow-GRPO. These results indicate that a more accurate dense reward significantly contributes to performance improvements. Conversely, a single-step ODE deviates substantially from the full denoising rollout, producing images that are far from being well-denoised. This deviation leads to less accurate rewards and, consequently, degraded overall performance.
>
> Following the Review 9Dzz' suggestion (Q2), we further conduct experiments with $n=t/C$, where $C$ is set to 3 and 2, respectively. As elaborated below, DenseGRPO with both $n=t/3$ and $n=2$ outperforms Flow-GRPO. Additionally, in the early training stage (200 GPU Hours), $n=t/2$ provides better results compared to $n=t$. However, in the later training stage, $n=t$ outperforms both $n=t/2$ and $n=t/3$ under the same training duration. This outcome can be attributed to the fact that a smaller $n$ accelerates the training process, but compromises the accuracy of reward predictions, ultimately limiting the performance ceiling. Consequently, with extended training durations, smaller values of $n$ fail to match the performance achieved with $n=t$.  Based on these findings, we recommend the setting $n=t$ for better performance.
>
> | Training Time (GPU Hours) |  200  |  400  |  600  |  800  |
> | ---- | :----: | :----: | :----: | :----: |
> | Flow-GRPO                 | 22.55 | 22.78 | 22.94 | 23.01 |
> | DenseGRPO($n=t/3$)        | 23.13 | 23.48 | 23.58 | 23.77 |
> | DenseGRPO($n=t/2$)        | **23.19** | 23.52 | 23.62 | 23.82 |
> | DenseGRPO($n=t$)          | 23.05 | **23.55** | **23.74** | **23.88** |
>
>
>
> > **Q6: The paper extensively mentions the results under different settings of noise level control 'a'. Since these results may be far from the definition of 'a', briefly mentioning the meaning of 'a' in the caption could avoid confusion.**
>
> Thank you for your detailed suggestion. The definition of 'a' is introduced in Paragraph 4, Sec. 3, which is the uniform noise level adopted by other methods, e.g, Flow-GRPO. We have supplemented the meaning of 'a' in the caption of Fig. 3.
>
>
>
>
> Thank you once again for your valuable comments. If you have any further questions or would like to discuss the work in greater detail, we are happy to provide clarification.

---

### Official Review · Reviewer_q5r8 · 2025-10-29

**Soundness:** 3
**Presentation:** 3
**Contribution:** 3
**Rating:** 6
**Confidence:** 3

**Summary:**

This paper proposes DenseGRPO, a reinforcement learning framework for aligning flow-matching models (e.g., text-to-image generation) with human preferences.
While previous GRPO-based approaches such as Flow-GRPO and DanceGRPO rely on a sparse reward that assigns a single trajectory-level reward to all denoising steps, DenseGRPO introduces a step-wise dense reward estimation method. Specifically, it leverages ODE-based denoising to obtain intermediate clean images and applies a reward model to compute per-step reward gains. In addition, the paper identifies a mismatch between uniform noise injection and the time-varying denoising process in existing methods, and proposes a reward-aware exploration space calibration scheme that adaptively adjusts timestep-specific stochasticity.
Empirical results on multiple text-to-image benchmarks show consistent improvements over Flow-GRPO and Flow-GRPO+CoCA.

**Strengths:**

- The paper addresses the well-known issue of sparse reward assignment in GRPO-based flow matching models and provides a conceptually simple fix.

- DenseGRPO demonstrates noticeable improvements across several text-to-image benchmarks and alignment metrics (e.g., +1.0 PickScore gain).

- Figures and algorithm descriptions are intuitive, especially the visualization of dense reward distributions and the adaptive exploration scheme.

**Weaknesses:**

- Conceptual Incrementality without a Deeper Credit Assignment Analysis

While the shift from sparse to dense reward seems natural, the paper does not provide a principled analysis of why ODE-based reward estimation captures per-step contribution more faithfully. The method assumes that ODE rollouts preserve semantic consistency, yet no theoretical or empirical verification supports this. Without a formal treatment of credit assignment or causality, the approach reads more as a heuristic refinement than a conceptual advance.

- Limited Perspective Beyond Flow Models

The method is tightly coupled with ODE-based denoising, making it unclear whether the same principle generalizes to other generative families (e.g., DDPM, consistency models). Without this, the claimed “general framework for dense reward” seems overstated.

- Unclear Attribution of Performance Gains

The performance boost could stem from either (a) the dense reward, (b) better noise scheduling, or (c) hyperparameter tuning. The ablation is not disentangled enough to isolate these effects. For example, the “uniform a=0.7” baseline in Fig.6(b) still benefits from DenseGRPO, which suggests confounding factors that aren’t fully analyzed.

**Questions:**

- Since dense rewards are computed using ODE rollouts while policy trajectories come from SDE sampling, how do the authors justify that the resulting reward estimates remain unbiased with respect to the actual policy distribution?
- Theoretical framing: Could the authors connect their dense reward to existing temporal credit-assignment frameworks (e.g., potential-based shaping or advantage decomposition)?
- Would DenseGRPO still hold if the base model were trained with a consistency loss or direct velocity regression instead of flow matching? Is the dense reward specific to ODE semantics?
- Is the reward gain $\Delta R_t = R_{t-1}-R_t$ guaranteed to reflect incremental improvement rather than numerical noise from the reward model? Has the variance of this difference been analyzed?

---

> ### Author Response · Authors · 2025-11-21
> **Response to Reviewer q5r8 (Part 1)**
>
> Thank you for your thoughtful and constructive feedback, which has been incredibly helpful in improving our work. Below, we respond to each of your comments in detail. We would like to point out that there are notable overlaps between **Weakness 1** and **Question 1**, as well as **Weakness 2** and **Question 3**. To ensure clarity and conciseness, we have addressed these points collectively and provided unified responses.
>
> > **W1 & Q1: [Conceptual Incrementality without a Deeper Credit Assignment Analysis] While the shift from sparse to dense reward seems natural, the paper does not provide a principled analysis of why ODE-based reward estimation captures per-step contribution more faithfully. The method assumes that ODE rollouts preserve semantic consistency, yet no theoretical or empirical verification supports this. Without a formal treatment of credit assignment or causality, the approach reads more as a heuristic refinement than a conceptual advance.**
>
> We would like to address your concern about the accuracy of DenseGRPO's dense reward as follows.
>
> In DenseGRPO, we estimate the step-wise dense reward by capturing the reward gain of the intermediate latents at a denoising step. Consequently, the accuracy of the dense reward inherently relies on the accuracy of the predicted latent reward at intermediate steps. A promising way for latent reward prediction is to establish rollouts to obtain the corresponding clean latents, assigning the reward of the clean latent as the latent reward. For example, ReFL[1] directly predicts the original $x_0$ from $x_t$ using a one-step SDE denoising in diffusion models and assigns the reward of $x_0$ as the latent reward of $x_t$. However, as demonstrated by Fig. 5 of [1], their latent reward fails to accurately reflect the terminal reward at high-noise timesteps. This is because only one-step SDE denosing deviates far from the complete rollouts, yielding out-of-distribution results and unreliable rewards. And employing multiple SDE denosing steps would inject additional stochasticity, disrupting the one-to-one mapping and impairing the semantic consistency. To tackle this, we propose an ODE-based approach for a more accurate estimation of the latent reward in flow matching models, where the deterministic nature of the ODE sampler ensures a one-to-one mapping while generating well-denoised outcomes. Once the current state is determined, all subsequent ODE denoising steps, along with the clean output, are fully defined without stochasticity. Thus, it is reliable and reasonable to assign the ODE-based terminal reward as the latent reward.
>
> To support this, we conduct an empirical verification by comparing the estimated latent reward with the terminal reward of the entire trajectory, as introduced in Sec. B.2 of the Appendix. The results presented in Fig. 8 show that the relative ranking of rewards across different samples remains consistent across all timesteps, demonstrating a minimal difference between the predicted latent rewards and the terminal trajectory rewards across both high- and low-noise stages. These findings validate the accuracy of the predicted reward, demonstrating the effectiveness of the ODE-based approach.
>
>
> [1] Jiazheng Xu, et al. Imagereward: Learning and evaluating human preferences for text-to-image generation. NIPS, 2023.

---

> ### Author Response · Authors · 2025-11-21
> **Response to Reviewer q5r8 (Part 2)**
>
> > **W2 & Q3: [Limited Perspective Beyond Flow Models] The method is tightly coupled with ODE-based denoising, making it unclear whether the same principle generalizes to other generative families (e.g., DDPM, consistency models). Without this, the claimed “general framework for dense reward” seems overstated.**
>
> Thank you for your valuable comment. The flow matching paradigm has demonstrated superior generative capabilities and has been extensively adopted in mainstream generative models, such as FLUX[1], Wan[2], and Qwen-Image[3]. Therefore, recent studies on human preference alignment have shown an increasing focus on flow matching models, including Flow-GRPO[4] and DanceGRPO[5]. In light of this trend, our work primarily concentrates on the flow matching model.
>
>
>
> While the ODE-based reward of DenseGRPO achieves significant advancement on flow matching models, it can generalize to other generative models as well. The key to this generalization is employing a deterministic denoising sampler to establish a one-to-one mapping between intermediate latent states and clean latents. This deterministic nature ensures the accurate prediction of latent rewards, which are critical for step-wise dense reward estimation. To validate this capability, we perform experiments on the diffusion model, as shown in Sec. B.3 of the Appendix. Specifically, similar to the approach applied to flow matching models, we conduct an experiment on diffusion model by utilizing SD 1.5[6] as the base model with an ODE sampler to predict $\hat{x}^{i}\_{t,0}$  from the $x^i_t $. The reward of $\hat{x}^{i}\_{t,0}$ is then assigned to that of $x^i_t$, and the reward gain is calculated as the dense reward. The result, presented in Fig. 9(c), indicates that the dense reward of DenseGRPO outperforms the sparse reward within Flow-GRPO, thereby demonstrating the accuracy and effectiveness of dense reward on diffusion models. These findings suggest that DenseGRPO is capable of generalizing to other generative models by employing a deterministic denoising sampler.
>
> [1] Forest Labs Black, et al. Flux.1 kontext: Flow matching for in-context image generation and editing in latent space. arXiv preprint, 2025.
>
> [2] Wan Team, et al. Wan: Open and advanced large-scale video generative models. arXiv preprint, 2025.
>
> [3] Chenfei Wu, et al. Qwen-image technical report. arXiv preprint, 2025.
>
> [4] Jie Liu, et al. Flow-grpo: Training flow matching models via online rl. arXiv preprint, 2025.
>
> [5] Zeyue Xue, et al. Dancegrpo: Unleashing grpo on visual generation. arXiv preprint, 2025.
>
> [6] Robin Rombach, et al. High-resolution image synthesis with latent diffusion models. CVPR, 2022.
>
>
> > **W3: [Unclear Attribution of Performance Gains] The performance boost could stem from either (a) the dense reward, (b) better noise scheduling, or (c) hyperparameter tuning. The ablation is not disentangled enough to isolate these effects. For example, the “uniform a=0.7” baseline in Fig.6(b) still benefits from DenseGRPO, which suggests confounding factors that aren’t fully analyzed.**
>
> We are grateful for your acknowledgment of our two key contributions: the dense reward and better noise scheduling. In Sec. 5.3, we have conducted ablation experiments to study their individual effects. To address your concerns more comprehensively, we provide a detailed discussion of the three factors you mentioned below.
>
> (a) **Dense reward.** As demonstrated by Flow-GRPO, $a=0.7$ is the best choice for uniform noise injection in the SDE sampler. Therefore, we employ $a=0.7$ as a baseline setting for noise injection. In Fig. 6(b), the DenseGRPO setting with $a=0.7$ differs from Flow-GRPO only in the choice between dense rewards and sparse rewards. The results show that DenseGRPO ($a=0.7$) also yields improved performance than Flow-GRPO, demonstrating the effectiveness of dense reward.
>
> (b) **Better noise scheduling.** When employing the proposed timestep-specific noise level $\psi(t)$ to replacing the uniform $a=0.7$, the achieved advancement shown in Fig. 6(b) validates the benefit of our better noise scheduling.
>
> (c) **Hyperparameter tuning.** To ensure a fair comparison in our experimental setup, all training configurations, such as the learning rate and other hyperparameters, were aligned with those of Flow-GRPO. Hence, the performance improvement is not attributed to hyperparameter tuning.
>
> In conclusion, both the introduction of the dense reward mechanism and the enhanced noise scheduling contribute significantly to the performance improvements, underscoring the superiority of our approach.

---

> ### Author Response · Authors · 2025-11-21
> **Response to Reviewer q5r8 (Part 3)**
>
> > **Q2: Theoretical framing: Could the authors connect their dense reward to existing temporal credit-assignment frameworks (e.g., potential-based shaping or advantage decomposition)?**
>
> Thank you for your insightful comment. Our DenseGRPO, featuring step-wise dense rewards, can be characterized as a temporal credit-assignment framework. More specifically, it belongs to the category of **forethought credit assignment**[1], where the future reward of an action is predicted immediately without the necessity of a full trajectory rollout. As mentioned in our paper, in the task of text-to-image generation, a typical implementation of forethought credit assignment is to train a process reward model[2] that predicts the future reward of a latent immediately, bypassing full trajectory rollouts. Similarly, DenseGRPO directly estimates the dense reward of a denoising step without assessing the final generated image. Compared to the process reward model, DenseGRPO offers two advantages: (1) it eliminates the need for additional specialized models and (2) it can seamlessly integrate with any established reward model.
>
>
> [1] Veronica Chelu, et al. Forethought and hindsight in credit assignment. NIPS, 2020.
>
> [2] Ziyi Zhang, et al. Confronting reward overoptimization for diffusion models: A perspective of inductive and primacy biases. ICML, 2024.
>
>
> > **Q4: Is the reward gain $\Delta R_t = R_{t-1}-R_t$ guaranteed to reflect incremental improvement rather than numerical noise from the reward model? Has the variance of this difference been analyzed?**
>
> Thank you for your thoughtful question. The reward noise is a common and inherent issue associated with existing reward models, and mitigating its effects often requires additional specialized designs. Nevertheless, compared with the trajectory-wise reward $R^i_0$ of Flow-GRPO, the step-wise reward $\Delta R^i_t = R^i_{t-1}-R^i_t$ within DenseGRPO does not amplify the impact of reward noise. Instead, the step-wise formulation provides a reliable reflection of the incremental improvements between steps.
>
> To elaborate, assume that the reward noise $\epsilon$ follows a Gaussian distribution $\mathcal{N}(0,\sigma^2)$ with a mean of 0 and a variance of $\sigma^2$, the predicted reward $R^i_t=\tilde{R}^i_t+\epsilon$ then follows the distribution $\mathcal{N}(\tilde{R}^i_t,\sigma^2)$, where $\tilde{R}^i_t$ represents the true, unbiased reward without noise. Accordingly, the advantage $\hat{A}\_t^i$ calculated in Flow-GRPO follows a t-distribution with $n−1$ degrees of freedom, denoted as $\hat{A}\_t^i \sim \mathcal{t}\_{n-1}$.In DenseGRPO, the step-wise reward $\Delta R^i\_t = R^i\_{t-1}-R^i\_t$ follows $\mathcal{N}(\tilde{R}^i\_{t-1} - \tilde{R}^i\_t,2 \sigma^2)$. And the corresponding advantage function $\hat{A}\_t^i$ also follows a t-distribution $\mathcal{t}\_{n-1}$, consistent with the advantage function in Flow-GRPO. This consistency ensures that the step-wise formulation in Dense-GRPO does not magnify the influence of reward noise, preserving a robust evaluation of the policy's performance.
>
> Additionally, as discussed in W1, the employed ODE-based approach in DenseGRPO offers an accurate estimation of the latent reward and the step-wise reward, reducing the noise during credit assignment. As a result, the reward gain $\Delta R^i_t = R^i_{t-1}-R^i_t$ can faithfully reflect the incremental improvement rather than reward noise.

---

> > ### Comment · Reviewer_q5r8 · 2025-11-26
> > **Official Comment by Reviewer q5r8**
> >
> > Thank you for your patient response in addressing my concerns. I will keep the score unchanged.

---

> > > ### Author Response · Authors · 2025-11-27
> > > **Thank you for your positive feedback**
> > >
> > > Thank you very much! We sincerely appreciate your acknowledgment of the value of our work and the constructive feedback you provided throughout this process. Your insights prompted us to further validate the effectiveness and robustness of the proposed method through additional experiments, and have greatly enhanced the quality of this manuscript.
> > >
> > > If you have any additional questions or would like further clarification on any point, please do not hesitate to let us know.

---

### Official Review · Reviewer_34g2 · 2025-10-31

**Soundness:** 3
**Presentation:** 3
**Contribution:** 3
**Rating:** 6
**Confidence:** 4

**Summary:**

This paper introduces DenseGRPO, which is a framework that enhances flow-matching based GRPO by introducing dense, step-wise rewards to better align human preferences with intermediate denoising contributions. It further calibrates the exploration space through a reward-aware, timestep-adaptive scheme, achieving more efficient and consistent alignment across benchmarks.

**Strengths:**

* The paper highlights the importance of **dense rewards** for effective **credit assignment** in reinforcement learning tasks. This is indeed a crucial factor that can significantly improve policy optimization stability and efficiency.
* Figure 6(c) provides informative results. It clearly demonstrates that the most straightforward approach of predicting the final $x_0$ in a single step to compute delta rewards performs even worse than the original Flow-GRPO, underscoring the necessity of proper dense reward design.

**Weaknesses:**

*  The proposed Dense-GRPO algorithm introduces many additional ODE denoising steps, which inevitably increase computation time. However, the paper lacks a comparison with Flow-GRPO under the same training-time horizontal axis to show efficiency differences.

*  Since the modified algorithm may alter KL-consumption behavior, it would be helpful to visualize how the KL loss evolves during training for both Flow-GRPO and Dense-GRPO.

*  The proposed Exploration Space Calibration module seems designed specifically for dense rewards. It would be valuable to clarify whether this technique can be directly applied to Flow-GRPO.

**Questions:**

What is the difference between the sparse reward setting in Figure 6(a) and the original Flow-GRPO, and why does it perform significantly better than the original version?

---

> ### Author Response · Authors · 2025-11-21
> **Response to Reviewer 34g2**
>
> We sincerely appreciate your insightful comments and efforts in reviewing our paper. We respond to each of your comments one by one in what follows.
>
> > **W1: The proposed Dense-GRPO algorithm introduces many additional ODE denoising steps, which inevitably increase computation time. However, the paper lacks a comparison with Flow-GRPO under the same training-time horizontal axis to show efficiency differences.**
>
> The comparison of the training process in terms of GPU hours is already presented in Fig. 6(c). The results show that, although the additional ODE denoising steps introduce higher computational costs, our Dense-GRPO with t-step ODEs achieves superior performance compared to Flow-GRPO under equivalent training time. This improvement is primarily attributed to the high-quality reward signals provided by DenseGRPO, which efficiently guide the learning process.
>
>
> > **W2: Since the modified algorithm may alter KL-consumption behavior, it would be helpful to visualize how the KL loss evolves during training for both Flow-GRPO and Dense-GRPO.**
>
> We have supplemented the visualization of the training curve for the KL loss in Sec. B.1 of the Appendix. This visualization shows that the KL loss of Dense-GRPO is slightly larger than that of Flow-GRPO. This difference can be attributed to the adoption of the timestep-specific noise level, which promotes a more diverse exploration space and consequently causes the model to deviate further from the original model.
>
>
> > **W3: The proposed Exploration Space Calibration module seems designed specifically for dense rewards. It would be valuable to clarify whether this technique can be directly applied to Flow-GRPO.**
>
> Thank you for your insightful comment. The results of Exploration Space Calibration, i.e., adjusted noise levels, can be directly applied to Flow-GRPO. However, the calibration process still requires dense rewards to evaluate whether the current noise level achieves a balanced exploration space.
>
> As shown below, we conduct an experiment by applying the timestep-specific noise level, calibrated using dense rewards, to Flow-GRPO. While using trajectory-wise sparse rewards for training, the calibrated timestep-specific noise level effectively facilitates a balanced exploration space for reinforcement learning.
>
>
> | Model / Training Step | Step 200 | Step 400 | Step 600 | Step 800 | Step 1000 |
> | -------- | :--------: | :--------: | :--------: | :--------: | :--------: |
> |  Flow-GRPO      | 22.16  | 22.44 | 22.59 | 22.72 | 22.78 |
> | **Flow-GRPO + Exploration Space Calibration** | **22.45** | **22.83** | **23.10** | **23.20** | **23.32** |
>
>
>
> > **Q1: What is the difference between the sparse reward setting in Figure 6(a) and the original Flow-GRPO, and why does it perform significantly better than the original version?**
>
> Thank you for your detailed review. In the setting "sparse reward", a trajectory-wise reward predicted by the ODE denoising rollout is assigned to each intermediate step, i.e., $R^i_t$ in Eq. 9. Apart from this difference, the other settings remain consistent with those of DenseGRPO. Since this reward is trajectory-wise, similar to Flow-GRPO, the term "sparse reward" was borrowed from the trajectory-wise sparse reward used in Flow-GRPO. To avoid the confusion this may cause, we have renamed this setting to "**Dense Reward (Baseline)**" for greater clarity, and updated Fig. 6(a) and Paragraph 1 of Sec. 5.3.
>
> Regarding the observed performance improvement, the "Dense Reward (Baseline)" setting assigns a more accurate reward to each intermediate step via the ODE denoising rollout, rather than relying on the delayed terminal reward assigned in Flow-GRPO. These results further underscore the critical role of dense reward in flow matching model alignment.

---

### Author Response · Authors · 2025-11-21
**General Response**

We sincerely thank all reviewers for their thoughtful and constructive reviews, which are helpful for the quality improvement of our paper. In the revised version, we have carefully addressed all concerns and suggestions raised by the reviewers. All revisions are highlighted in red for clarity, while newly introduced sections are distinguished by red titles only. The key updates are summarized below:

- **Implementation Detail (Appendix Sec. A):** We include additional implementation details regarding DenseGRPO to improve transparency and reproducibility.
- **Training Curve of KL Loss (Appendix Sec. B.1):** We visualize the training curve for the KL loss to offer better insights into the optimization dynamics.
- **Accuracy of DenseGRPO's Reward (Appendix Sec. B.2):** We evaluate the accuracy of DenseGRPO's dense reward to demonstrate its reliability and effectiveness.
- **Additional Experiments (Appendix Sec. B.3):** We conduct additional experiments on FLUX.1-Dev, high-resolution tasks, and diffusion models to further validate DenseGRPO's effectiveness and generalization potential.
- **Reward Hacking Analysis (Appendix Sec. B.4):** We include a detailed analysis of reward hacking, covering qualitative cases, underlying causes, and potential solutions to mitigate observed challenges.
- **Statement:** We add the Ethics Statement and Reproducibility Statement at the end of the main text (before references). Additionally, the usage of LLMs is presented in Appendix Sec. C.
- **Clarity Improvement:** We make some minor revisions to improve the clarity of the manuscript.

We provide detailed responses to each reviewer below and remain open to any further questions or discussions. Thank you again for your valuable feedback and support in improving this work.

---

### Author Response · Authors · 2025-12-03
**Quick Summary for Area Chair**

We recognize the severe impact of the information leak incident on Nov 27 and sincerely appreciate your time and effort for our work. Although the leak incident disrupted the rebuttal process, we are fortunate that the reviewers actively engaged in discussions prior to the leak. We are encouraged that our rebuttal successfully addressed their major concerns, and **all the reviewers provided positive evaluations on our work**. To respect your time, we provide a concise summary of our rebuttal below:

### _**Rating: 6 6 6 4 $ \rightarrow $  6 6 6 6 (before Nov 26, prior to the leak incident)**_

Prior to the information leak incident on Nov 27, our rebuttal was thoroughly examined by the reviewers and succeeded in addressing most of their concerns, leading to a rating improvement from 6, 6, 6, 4 to 6, 6, 6, 6.

**Critical Timeline：**
* `Nov 12`: Initial Rating: 6, 6, 6, 4.
* `Nov 21`: Rebuttal submitted.
* `Nov 22, 11:47 UTC`: Reviewer Fc1V noted that our rebuttal resolves many of the previous concerns and **raised the score** for Presentation to 3.
* `Nov 25, 12:33 UTC`: Reviewer 9Dzz **raised score** (4 $ \rightarrow $ 6) after the concerns have been addressed.
* `Nov 26, 11:53 UTC`: Reviewer q5r8 kept the positive score.
* `Nov 27 (PM) UTC`: The information leak occurred (approx. **2 days** after our final score raise).


### _**Summary of Concern and Rebuttal**_

All reviewers expressed their endorsement of our work, and the concerns raised were minimal. We summarize their main concerns as follows.


* **Reviewer 34g2 (Rating 6)** &  **Reviewer q5r8 (Rating 6)** & **Reviewer Fc1V (Rating 6)**

	* **Accuracy of DenseGRPO's Reward:** We evaluate the accuracy of DenseGRPO's dense reward, demonstrating its reliability and effectiveness.
	* **Generalization:** We carry out additional experiments on Flow-GRPO, FLUX.1-Dev, high-resolution tasks, and diffusion models to further validate DenseGRPO's effectiveness and generalization potential.
	* **Deeper Reward Hacking Analysis:** We provide a comprehensive analysis of reward hacking, including qualitative examples, an examination of underlying causes, and potential mitigation strategies for the identified challenges.



* **Reviewer 9Dzz (Rating 4 $\rightarrow$ 6)**
> "_**Most of my concerns have been addressed, so I raise my score to 6.**_"

	* **Difference between Reward in Traditional RL and Dense Reward:** We provide a detailed clarification of this distinction, offering a comprehensive understanding of our reward formulation.
	* **The Goal of Exploration Space Calibration:** We present more thorough explanations to address and clarify any potential confusion.


Thank you once again for devoting your time and effort to evaluating our work.

---

### Meta-Review · Area_Chair_82iL · 2026-01-05

**Summary:**

The reviewers' main concerns are:

1. Many additional ODE steps increases computation time (reviewer 34g2).

2. About lack of deeper credit assignment analysis (reviewer q5r8).

3. Unclear attribution of performance gains (reviewer q5r8).

4. Limited analysis of reward hacking (reviewer Fc1V).

5. Generalization questions from reviewer Fc1V.

6. Several notational typos and no code repo (reviewer 9Dzz)

**Reviewer Concerns:**

I have read the paper carefully. After the rebuttal, the concerns 1,2,3,5,6 above are mostly addressed properly. The authors also provide analysis of reward hacking for concern 4. As is also shown in Table 1, in Compositional Image Generation and Visual Text Rendering, the proposed method does not compare favorably compared to the baseline models in terms of image quality. This shall be more detailedly discussed in the paper. Also, the authors are strongly encouraged to open-source the code for reproducibility.

**Reviewer Scores:**

After the rebuttal, I think the reviewer will have the score of 6,6,6,6.

---

### Decision · Program_Chairs · 2026-01-26

Accept (Poster)